# Effects of route of administration on oxytocin-induced changes in regional cerebral blood flow in humans

D.A. Martins[1], N. Mazibuko[1], F. Zelaya[1], S. Vasilakopoulou[1], J. Loveridge[1], A. Oates[2], S. Maltezos[3,4], M. Mehta [1], S. Wastling[1,5,6], M. Howard [1], G. McAlonan[7], D. Murphy [7], S.C.R. Williams[1], A. Fotopoulou[8], U. Schuschnig[9] & Y. Paloyelis [1✉]

Could nose-to-brain pathways mediate the effects of peptides such as oxytocin (OT) on brain physiology when delivered intranasally? We address this question by contrasting two methods of intranasal administration (a standard nasal spray, and a nebulizer expected to improve OT deposition in nasal areas putatively involved in direct nose-to-brain transport) to intravenous administration in terms of effects on regional cerebral blood flow during two hours post-dosing. We demonstrate that OT-induced decreases in amygdala perfusion, a key hub of the OT central circuitry, are explained entirely by OT increases in systemic circulation following both intranasal and intravenous OT administration. Yet we also provide robust evidence confirming the validity of the intranasal route to target specific brain regions. Our work has important translational implications and demonstrates the need to carefully consider the method of administration in our efforts to engage specific central oxytocinergic targets for the treatment of neuropsychiatric disorders.

[1] Department of Neuroimaging, Institute of Psychiatry, Psychology and Neuroscience, King's College London, London, UK. [2] South London and Maudsley NHS Foundation Trust, London, UK. [3] Adult Autism and ADHD Service, South London and Maudsley NHS Foundation Trust, London, UK. [4] Institute of Psychiatry, Psychology and Neuroscience, King's College London, London, UK. [5] Department of Brain Repair and Rehabilitation, Institute of Neurology, University College London, London, UK. [6] Lysholm Department of Neuroradiology, National Hospital for Neurology and Neurosurgery, London, UK. [7] Department of Forensic and Neurodevelopmental Science (SM), Institute of Psychiatry, Psychology and Neuroscience, King's College London, London, UK. [8] Department of Clinical, Educational and Health Psychology, University College London, London, UK. [9] PARI GmbH, Gräfelfing, Germany. ✉email: yannis.paloyelis@kcl.ac.uk

Over the past decade, robust evidence from preclinical studies has demonstrated the importance of the central OT system in the development[1] and regulation of social behaviour[2,3], the modulation of pain processing[4], feeding behaviour[5] and neuroinflammation after brain ischemia[6]. Harnessing the central OT system has been identified as a promising strategy in translational neuroscience for the development of targeted pharmacological interventions to improve outcome in several conditions currently lacking efficacious treatments (e.g. autism spectrum disorder[7], schizophrenia[8], migrain[9], stroke[6], obesity[10], Prader-Willi[11]).

Human studies almost exclusively target the central OT system by administering synthetic OT using nasal sprays, despite a lack of understanding of the mechanisms underpinning its pharmacodynamic effects. OT is a hydrophilic cyclic nonapeptide and is unable to cross the blood-brain barrier (BBB) in significant amounts[12]. The intranasal administration of OT has been favoured under the assumption that once in the nasal cavity, OT can reach the brain directly, bypassing the BBB[13]. Two main mechanisms have been suggested to underpin the direct nose-to-brain transport[14]. The first mechanism postulates internalization of OT into olfactory or trigeminal neurons innervating the posterior and middle areas of the nasal cavity, followed by axonal transport and central exocytosis. However, this mechanism would be slow[15] and therefore could not account for the central and behavioural effects we observe within 15–60 min post-dosing in humans[16]. The second mechanism postulates that OT reaches the cerebrospinal fluid (CSF) and brain parenchyma via passive diffusion through perineural clefts in the nasal epithelium, which provide a gap in the blood-brain barrier[17]. While some animal work is consistent with the existence of the second mechanism[18], there is a lack of robust evidence to support the existence of nose-to-brain transport in humans[15,19].

The lack of clarity regarding the mechanisms mediating the effects of intranasal OT in humans and the inconsistent results in existing studies and clinical trials using intranasal sprays to deliver OT[20] have raised questions about the validity of the intranasal route to administer OT to the brain. While very small amounts of intranasally administered OT have been reported to reach the CSF[15], peripheral concentrations in the blood are also increased to supraphysiologic levels. The increase in plasma OT levels unavoidably engages OT receptors expressed throughout the body[15] and may impact indirectly on brain function and behaviour. It is also possible that the small amount of synthetic OT that crosses the BBB from systemic circulation[15] may be sufficient to induce functional effects in the brain, either by directly activating receptors in the brain or by stimulating OT autoreceptors on OT-synthesizing hypothalamic neurons to induce the release endogenous OT[15]. These mechanisms might account for the effects of peripherally administered OT (e.g. intravenous infusion) on behaviour[21–24].

Intranasal delivery allows for fast absorption of small molecules into the peripheral circulation[25,26], at the cost of poor and unreliable control of the amount of the drug absorbed[25,26]. Therefore, to achieve significant translational advances, we need to confirm whether nasal pathways can be used to target the central OT system and whether they offer any advantage in relation to alternative methods.

The absence of a selective radiolabelled OT ligand in humans makes it impossible to directly examine the central penetration and distribution of synthetic OT after intranasal administration. An alternative strategy is to quantify whole-brain functional effects after OT administration. We have previously demonstrated the sensitivity of arterial spin labelling (ASL) magnetic resonance imaging (MRI) in quantifying changes in brain's physiology after intranasal OT administration[16], as reflected in changes in regional cerebral blood flow (rCBF) at rest, which provide a quantitative, non-invasive pharmacodynamic marker of the effects of acute doses of psychoactive drugs[27,28], with high-spatial resolution and excellent temporal reproducibility[29]. As a result of neuro-vascular coupling, changes in rCBF are likely to reflect changes in neuronal activity[30], and they capture relevant differential neurotransmitter activity of neurochemical systems[16,31].

In this study, we use ASL MRI to investigate and compare the changes in rCBF that follow intranasal and intravenous OT administration over 104 min post-dosing (Fig. 1). The use of an intravenous comparator can illuminate whether intranasal OT-induced changes in brain perfusion in humans result from concomitant increases in systemic OT circulation. Standard nasal sprays were not designed to maximise deposition in the olfactory and respiratory epithelia that is thought to mediate nose-to-brain transport[32]. Therefore, alongside a standard nasal spray, we used a nasal administration method (*PARI SINUS* nebuliser) that combines the production of small size droplets with vibration to maximize deposition in upper and posterior regions of the nasal cavity where the nose-to-brain transport putatively occurs[33].

We reasoned that if intranasal administration represents a privileged route for the central delivery of OT, then intranasal OT-induced changes in rCBF in brain regions typically associated with the effects of OT in the brain (e.g. the amygdala)[34–36] should not be explained by increases in plasmatic OT achieved after OT intravenous infusion. Furthermore, if posterior regions of the nasal cavity are involved in nose-to-brain transport, then using a device that can increase deposition in these areas may result in a more robust pattern of changes in rCBF when compared to OT administration with a standard nasal spray. We demonstrate that OT-induced decreases in amygdala perfusion, a key hub of the OT central circuitry, are explained entirely by OT increases in systemic circulation following both intranasal and intravenous OT administration. Yet we also provide robust evidence confirming the validity of the intranasal route to target specific brain regions.

## Results

**Global CBF and subjective state ratings**. We observed a linear decrease over time in participants' global CBF level and levels of alertness and excitement (Main effect of time-interval); however, there was no significant main effect of treatment or time-interval × treatment interaction (Supplementary Figs. 1 and 2 and Supplementary Table 1). We did not observe any significant correlation between changes in global CBF and ratings of alertness or excitement over time (Supplementary Fig. 3).

**Whole-brain univariate analyses**. We first computed a flexible factorial model to investigate treatment, time-interval and time-interval × treatment effects on rCBF. We did not observe a significant main effect of treatment; however, we found a significant main effect of time-interval on rCBF in several clusters across the brain, likely reflecting decreases in alertness and attention. We observed a significant treatment × time-interval interaction in three clusters. These clusters spanned: (1) the left superior and middle frontal gyri and the anterior cingulate gyrus; (2) the right occipital gyrus, cerebellum, lingual and fusiform gyri, calcarine cortex, cuneus and inferior temporal gyrus; (3) the left putamen, caudate nucleus, insula, amygdala, parahippocampal gyrus, rectus gyrus and medial orbitofrontal cortex (Fig. 2 and Table 1).

We followed up the flexible factorial model with an exhaustive series of paired *t*-tests at each time interval to investigate the direction of potential OT-induced changes in rCBF specifically for each treatment route (compared to placebo). Results are

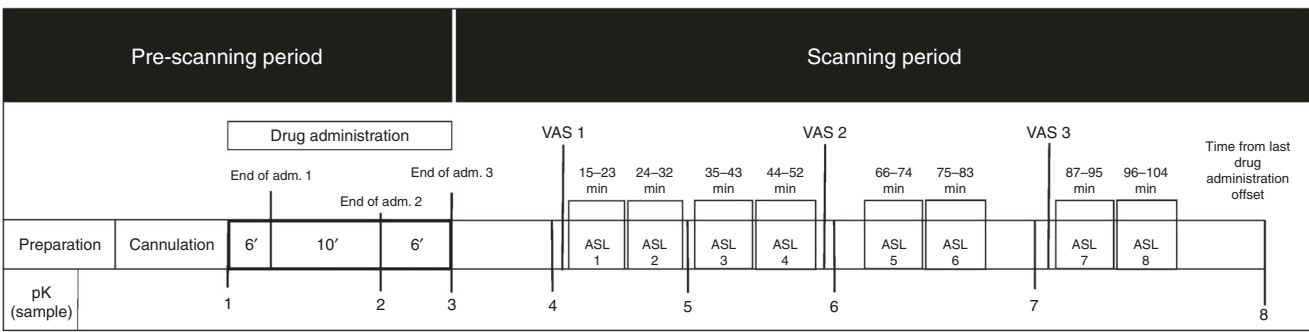

**Fig. 1 Study design, drug administration and blood sampling procedures.** Upper panel—Drug allocation scheme: In each session, participants received treatment via all three administration routes, in one of two fixed sequences: either nebulizer/intravenous infusion/standard nasal spray, or standard nasal spray/intravenous infusion/nebulizer. In 3 out of 4 sessions only one route of administration contained the active drug; in the fourth session, all routes delivered placebo or saline. Unbeknown to the participants, the first treatment administration method in each session always contained placebo, while intranasal (spray or nebulizer) OT was only delivered with the third treatment administration. The second administration was an intravenous nfusion of either saline or oxytocin. Lower panel—Study protocol: After drug administration, participants were guided to the MRI scanner, where eight pulsed continuous arterial spin labelling (ASL) scans (each lasting ~8 min) where acquired, spanning 15–104 min after last drug administration offset. We assessed participants' levels of alertness and excitement using visual analogue scales (VAS) at three different timepoints during the scanning session to evaluate subjective drug effects across time. We collected plasma samples at baseline and at five timepoints post-dosing to measure changes in the concentration of OT and perform pharmacokinetics analysis. (NB Nebulizer, IV Intravenous, pK Pharmacokinetics).

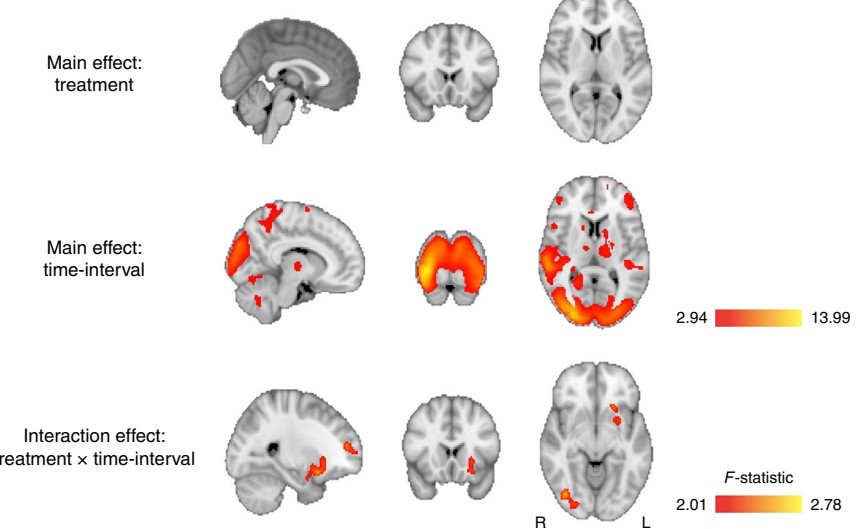

**Fig. 2 Whole-brain flexible factorial univariate analysis.** We firstly implemented an analysis of covariance design, controlling for global effects on CBF, using a flexible factorial model in SPM12 where we specified the factors Subjects, Treatment, and Time interval. We present the results of an F test to investigate the main effects of treatment, i.e. identify brain regions where any treatment induced persistent changes in rCBF (regardless of direction) across time (first row), and time-interval, i.e. identify brain regions where time induced persistent changes in rCBF (regardless of direction) across treatments (second row). We also present the results of an F test to investigate the interaction of Treatment × Time-interval i.e. identify brain regions showing specific treatment-induced changes in rCBF as a function of time interval, regardless of direction (third row). We conducted cluster-level inference, reporting clusters significant at $p < 0.05$ FWE-corrected (cluster-forming threshold: $p < 0.005$, uncorrected). Images are shown as F-statistics in radiological convention.

**Table 1 Whole-brain univariate flexible factorial analysis.**

| Cluster description | Hemisphere | K | P_fwe | Peak coordinates | | | Description |
|---|---|---|---|---|---|---|---|
| | | | | x | y | z | |
| Main effect: method | | | | | | | |
| No cluster survived correction | | | | | | | |
| Main effect: time interval | | | | | | | |
| Cluster 1 | Bilateral | 295660 | <0.001 | 38 | −84 | −8 | Right inferior occipital gyrus |
| | | | | 30 | −90 | −2 | |
| | | | | 28 | −82 | 14 | Right superior occipital gyrus |
| Cluster 2 | Left | 2127 | <0.001 | −14 | −18 | −4 | Left ventral diencephalon |
| | | | | −30 | −2 | −6 | Left putamen |
| | | | | −18 | −6 | 26 | Left caudate |
| Cluster 3 | Left | 2271 | <0.001 | −48 | −52 | 40 | Left angular gyrus |
| | | | | −56 | −34 | 18 | Left planum temporale |
| | | | | −42 | −32 | 12 | |
| Cluster 4 | Left | 854 | <0.001 | −46 | −70 | −42 | Left cerebellum |
| | | | | −18 | −70 | −36 | |
| | | | | −30 | −64 | −38 | |
| Cluster 5 | Bilateral | 2389 | <0.001 | −12 | 28 | 50 | Left superior frontal gyrus |
| | | | | −40 | 16 | 52 | Left middle frontal gyrus |
| | | | | −42 | 24 | 50 | |
| Cluster 6 | Right | 435 | 0.022 | 48 | 4 | 42 | Right precentral gyrus |
| | | | | 50 | −2 | 42 | |
| Cluster 7 | Right | 508 | 0.010 | 36 | 52 | −14 | Right lateral orbital gyrus |
| | | | | 44 | 44 | −14 | |
| | | | | 46 | 44 | −4 | Right inferior frontal gyrus (triangular part) |
| Interaction: method × time interval | | | | | | | |
| Cluster 1: superior and middle frontal gyrus, anterior cingulate | Left | 341 | 0.040 | −18 | 46 | 18 | Left superior frontal gyrus |
| | | | | −26 | 56 | 8 | |
| | | | | −14 | 56 | 4 | |
| Cluster 2: inferior, superior and middle occipital gyrus, cerebellum, lingual gyrus, fusiform gyrus, calcarine cortex, cuneus, inferior temporal gyrus | Right | 905 | <0.001 | 38 | −78 | −12 | Right inferior occipital gyrus |
| | | | | 42 | −74 | −26 | Right cerebellum |
| | | | | 16 | −86 | −2 | Right calcarine cortex |
| Cluster 3: putamen, insula, amygdala, parahippocampal gyrus, caudate, olfactory region, rectus gyrus and medial orbitofrontal cortex | Left | 407 | 0.016 | −24 | 12 | −14 | Left putamen |
| | | | | −20 | 26 | 2 | Left caudate |
| | | | | −20 | 24 | −8 | |

Clusters showing a significant main effect of treatment, time or treatment × time-interval interaction in rCBF in the 15–104 min post-administration period, as identified in F contrasts (not capturing the direction of the change in rCBF). Global CBF was used as a nuisance variable. We conducted cluster-level inference, reporting clusters significant at $p < 0.05$ FWE-corrected (cluster-forming threshold: $p < 0.005$, uncorrected).

summarized in Fig. 3 and Tables 2–4. Given the novel and exploratory nature of these analyses we report the exact FWE-corrected *P*-values for significant clusters without further adjustment for the number of paired *t*-tests using conservative Bonferroni correction, and denote in Tables 2–4 with an asterisk those few clusters with $P_{FWE} > 0.002$ ($P = 0.05/24$ tests = 0.002) following Bonferroni adjustment.

Compared to placebo, synthetic OT when administered with the standard nasal spray decreased rCBF at 24–32 min post-dosing in a cluster extending over the left amygdala, left insula, left parahippocampal gyrus and hippocampus, and left temporal pole, and at 87–95 min post-dosing in two clusters, one including the anterior cingulate and the right superior/medial frontal gyri and another cluster spanning over the brainstem and the right cerebellum. We further observed significant increases in rCBF at 15–23 min post-dosing, in one cluster spanning the left superior/middle frontal gyri, supplementary motor area and the precentral gyrus, at 35–43 min post-dosing over a similar cluster (but

restricted to the left middle/inferior frontal and precentral gyri), and at 87–95 mins post-dosing in two clusters, one spanning the superior/middle temporal gyri, the posterior insula and the postcentral gyrus and another one involving the superior/inferior parietal lobes, the postcentral and precentral gyri and the precuneus (all in the left hemisphere) (Table 2).

Intravenous OT decreased rCBF over the same clusters and time-intervals as for the comparison between standard nasal sprays vs. placebo, but only in 2 time-intervals. Specifically, we observed significant decreases in rCBF at 15–23 min post-dosing in a cluster spanning the amygdala, insula, parahippocampal gyrus and globus pallidum (all left hemisphere), and at 87–95 min post-dosing in a cluster extending over the anterior cingulate, the superior frontal gyrus and the orbitofrontal cortex bilaterally (Table 3).

The administration of intranasal OT with the nebulizer resulted in a different pattern of changes compared to standard nasal spray. Specifically, we observed decreases in rCBF at 15–23 min

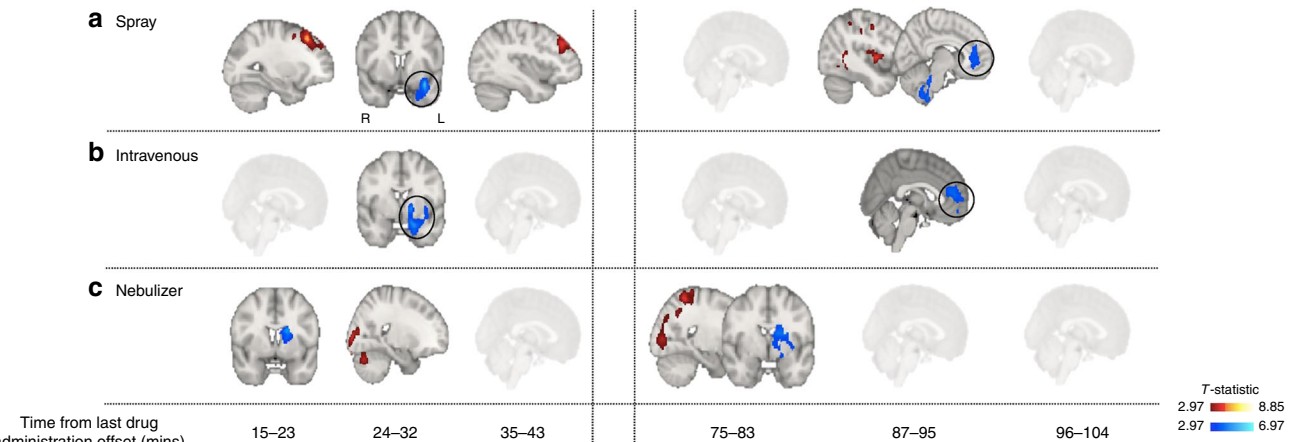

**Fig. 3 Whole-brain characterisation of the changes in rCBF associated with each method of administration.** For each time-point, we compared CBF maps between treatment and placebo using Paired T-contrasts, accounting for global CBF as a nuisance variable. We conducted cluster-level inference, reporting clusters significant at $p < 0.05$ FWE-corrected (cluster-forming threshold: $p < 0.005$, uncorrected). Images are shown as $t$-statistics in radiological convention. Blue and red indicate decreases and increases in rCBF, respectively. The first row (Panel **a**) refers to rCBF changes associated with the nasal spray, the second (Panel **b**) with the intravenous administration and the third one (Panel **c**) with the PARI SINUS nebulizer.

| Table 2 Paired $t$-contrasts comparing standard spray OT to placebo/saline. | | | | | | | |
|---|---|---|---|---|---|---|---|
| **Cluster description** | **Hemisphere** | **K** | **P$_{fwe}$** | **Peak coordinates** | | | **Description** |
| | | | | **x** | **y** | **z** | |
| Spray > placebo (15–23 min) | | | | | | | |
| Cluster 1: superior/middle frontal gyrus, supplementary motor area, precentral gyrus | Left | 1500 | <0.001 | −24 | 24 | 64 | Left superior/middle frontal gyrus |
| | | | | −26 | 28 | 46 | |
| | | | | −24 | 16 | 68 | |
| Spray < placebo (24–32 min) | | | | | | | |
| Cluster 2: parahipocampal gyrus, temporal pole, amygdala, insula, hippocampus | Left | 1295 | <0.001 | −34 | 8 | −20 | Left temporal pole |
| | | | | −16 | −6 | −34 | Left enthorinal area |
| | | | | −18 | −6 | −26 | Left enthorinal area |
| Spray > placebo (35–43 min) | | | | | | | |
| Cluster 3: middle and inferior frontal gyrus, precentral gyrus | Left | 999 | <0.001 | −42 | 40 | 34 | Left middle frontal gyrus |
| | | | | −54 | 20 | 38 | |
| | | | | −42 | 46 | 28 | |
| Spray > placebo (87–95 min) | | | | | | | |
| Cluster 4: superior and middle temporal gyrus, insula, postcentral gyrus | Left | 1848 | <0.001 | −56 | −46 | 2 | Left middle/superior temporal gyrus |
| | | | | −66 | −20 | 30 | Left postcentral/supramarginal gyrus |
| | | | | −60 | −46 | −8 | Left middle temporal gyrus |
| Cluster 5: superior and inferior parietal lobe, postcentral and precentral gyrus, precuneus | Left | 1315 | <0.001 | −18 | −54 | 52 | Left superior parietal lobe |
| | | | | −36 | −44 | 52 | Left superior parietal lobe/ supramarginal gyrus |
| | | | | −24 | −20 | 66 | Left precentral gyrus |
| Spray < placebo (87–95 min) | | | | | | | |
| Cluster 6: anterior cingulate, right superior and medial frontal gyrus | Right | 1350 | <0.001 | 24 | 46 | 12 | Right middle frontal gyrus |
| | Right | | | 16 | 46 | 12 | Right superior frontal gyrus |
| | Bilateral | | | −2 | 36 | 28 | Anterior cingulate gyrus |
| Cluster 7: brainstem and cerebellum* | Bilateral | 591 | 0.01 | 6 | −40 | −56 | Brainstem |
| | Right | | | 14 | −48 | −56 | Right cerebellum |
| | Bilateral | | | 6 | −38 | −32 | Brainstem |

We compared CBF maps following standard spray OT and placebo administration using paired $t$-contrasts at each time-interval (to capture the direction of potential rCBF changes), controlling for global CBF as a nuisance variable. We conducted cluster-level inference, reporting clusters significant at $p < 0.05$ FWE-corrected (cluster-forming threshold: $p < 0.005$, uncorrected). Clusters that do not survive correction for multiple testing following adjustment of the $P$-value using Bonferroni correction for the 24 paired-$t$-tests performed are denoted by an asterisk (*) (P$_{adjusted}$ = 0.05/24 = 0.002).

post-dosing in a cluster spanning the left caudate, left putamen and pallidum, and at 75–83 min post-dosing in a cluster extending over the left caudate, putamen, pallidum, thalamus, amygdala, hippocampus, olfactory region and the insula. We also observed increases in rCBF at 24–32 min post-dosing in two clusters, one spanning the right superior/middle/inferior occipital gyri, the calcarine sulcus and the cuneus bilaterally, and the other one

the right cerebellum, and at 75–83 min post-dosing in a cluster spanning the postcentral gyrus, the superior/middle/inferior occipital gyri, the superior parietal gyrus, the inferior/middle temporal gyri, the precuneus, the calcarine sulcus and the cuneus, all in the right hemisphere. Accounting for plasma OT AUC had no effect on the changes in rCBF observed following administration with the nebulizer (Table 4).

**Table 3 Paired *t*-contrasts comparing intravenous OT administration to placebo/saline.**

| Cluster description | Hemisphere | K | P_FWE | Peak Coordinates | | | Description |
|---|---|---|---|---|---|---|---|
| | | | | x | y | z | |
| **Intravenous < placebo (24–32 min)** | | | | | | | |
| Cluster 1: parahippocampal gyrus, pallidum, amygdala, insula | Left | 1159 | <0.001 | −12 | 4 | −2 | Left pallidum |
| | | | | −16 | 2 | −20 | Left enthorinal area/amygdala |
| | | | | −38 | 2 | −10 | Left anterior insula |
| **Intravenous < placebo (87–95 min)** | | | | | | | |
| Cluster 2: anterior cingulate, superior frontal gyrus, orbitofrontal cortex | Bilateral | 1179 | <0.001 | 6 | 42 | 16 | Right anterior cingulate |
| | | | | 14 | 62 | 12 | Right superior frontal gyrus |
| | | | | 10 | 56 | −6 | Right superior frontal gyrus |

We compared CBF maps following intravenously administered OT and placebo/saline using paired *t*-contrasts at each time-interval (to capture the direction of potential rCBF changes), controlling for global CBF as a nuisance variable. We conducted cluster-level inference, reporting clusters significant at *p* < 0.05 FWE-corrected (cluster-forming threshold: *p* < 0.005, uncorrected). Clusters that do not survive correction for multiple testing following adjustment of the *P*-value using Bonferroni correction for the 24 paired-*t*-tests performed are denoted by an asterisk (*) (P_adjusted = 0.05/24 = 0.002).

**Table 4 Paired *t*-contrasts comparing OT administered with the nebulizer to placebo/saline.**

| Cluster description | Hemisphere | K | P_fwe | Peak coordinates | | | Description |
|---|---|---|---|---|---|---|---|
| | | | | x | y | z | |
| **Nebulizer < placebo (15–23 min)** | | | | | | | |
| Cluster 1: caudate, putamen, pallidum* | Left | 434 | 0.037 | −18 | 0 | 20 | Left caudate |
| | | | | −12 | 8 | 8 | |
| | | | | −6 | 2 | 16 | |
| **Nebulizer > placebo (24–32 min)** | | | | | | | |
| Cluster 2: superior, middle and inferior occipital gyrus, calcarine sulcus, cuneus | Bilateral | 1154 | <0.001 | 52 | −74 | −8 | Right inferior occipital gyrus |
| | | | | 38 | −92 | −2 | |
| | | | | 40 | −88 | −14 | |
| Cluster 3: cerebellum* | Right | 457 | 0.033 | 26 | −72 | −28 | Right cerebellum |
| | | | | 12 | −82 | −28 | |
| | | | | 4 | −86 | −38 | |
| **Nebulizer > placebo (75–83 min)** | | | | | | | |
| Cluster 4: postcentral gyrus, superior, middle and inferior occipital gyrus, superior parietal gyrus, inferior and middle temporal gyrus, precuneus, calcarine sulcus, cuneus | Right | 2546 | <0.001 | 56 | −66 | −16 | Right inferior temporal gyrus |
| | | | | 28 | −54 | 72 | Right Superior Parietal Lobule |
| | | | | 48 | −78 | −16 | Right inferior occipital gyrus |
| **Nebulizer < placebo (75–83 min)** | | | | | | | |
| Cluster 5: caudate, putamen, pallidum, thalamus, amygdala, hippocampus, olfactory region, insula | Left | 1229 | <0.001 | −14 | 0 | 18 | Left caudate |
| | | | | −12 | −14 | −10 | Left ventral diencephalon |
| | | | | −6 | −6 | 4 | Left thalamus |

We compared CBF maps following OT administered with the nebulizer to placebo/saline using paired *t*-contrasts at each time-interval (to capture the direction of potential rCBF changes), controlling for global CBF as a nuisance variable. We conducted cluster-level inference, reporting clusters significant at *p* < 0.05 FWE-corrected (cluster-forming threshold: *p* < 0.005, uncorrected). Clusters that do not survive correction for multiple testing following adjustment of the *P*-value using Bonferroni correction for the 24 paired-*t*-tests performed are denoted by an asterisk (*) (P_adjusted = 0.05/24 = 0.002).

Decreases in rCBF observed after standard nasal spray and intravenous OT administration at 24–32 and 87–95 min post-dosing overlap anatomically to a substantial extent (Fig. 4). We thus followed up with a direct comparison of standard nasal spray vs. intravenous OT administration using paired sample *t*-tests for these time-intervals; we did not observe any significant differences in rCBF between the two administration methods. Individual differences in OT concentration (AUC over 120 min post-dosing) in plasma post-dosing were negatively correlated with OT-induced changes in rCBF in the two spatially over-lapping clusters showing significant decreases in rCBF in the comparisons of nasal spray OT vs. placebo and intravenous OT vs. saline (Fig. 5), but not with any other OT-induced changes, irrespective of method of administration, time-interval or direction of effect (Supplementary Table 3). Accounting for

plasma OT AUC in the paired *t*-test for the standard nasal spray vs. placebo and the intravenous OT vs. saline comparisons eliminated all significant decreases in rCBF that were observed for each of these administration methods. However, accounting for plasma OT AUC had no effect on the changes in rCBF uniquely observed in the standard nasal spray vs. placebo comparison.

**Comparison of pharmacokinetic profiles among treatments.** OT reached peak plasma concentration at the end of dosing when administered intravenously or via nasal spray, and by ~15 min post-dosing when administered with the nebulizer (Fig. 6a). Intravenous peak plasma OT concentrations ($C_{max}$) were significantly higher than either intranasal administration method, while $C_{max}$ did not differ between the nasal administration

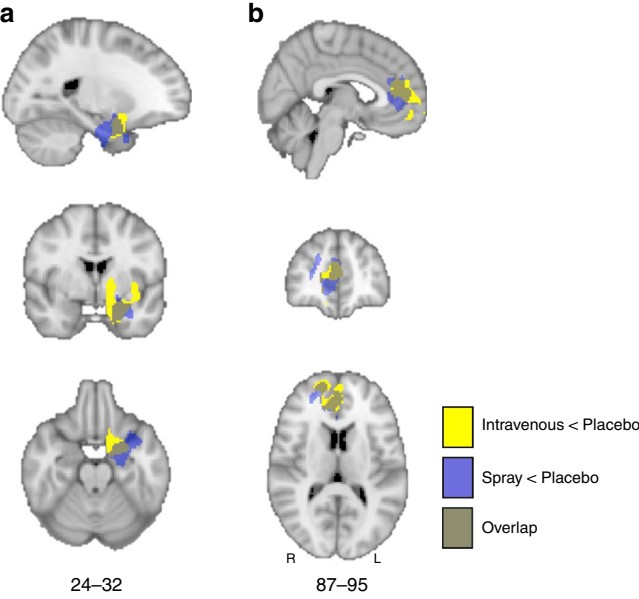

Fig. 4 Intravenous and Spray OT decreased rCBF in areas overlapping anatomically. Panel **a** refers to rCBF changes identified at 24–32 min post-dosing; Panel **b** refers to rCBF changes identified at 87–95 min post-dosing. In yellow we present the voxels where OT intravenous administration decreased rCBF at these timepoints, while blue refers to the voxels where OT administered with a spray decreased rCBF at the same timepoints. We conducted cluster-level inference, reporting clusters significant at $p < 0.05$ FWE-corrected (cluster-forming threshold: $p < 0.005$, uncorrected). Images are shown as binary clusters in radiological convention.

methods (Supplementary Table 2) (Repeated measures One-way ANOVA: $F_{(1.38,22.02)} = 92.39$, $p < 0.0001$, $\eta^2_p = 0.815$, post-hoc tests: Spray vs Nebulizer—$p = 0.972$; Spray vs Intravenous—$p < 0.0001$; Nebulizer vs Intravenous—$p < 0.0001$). As intended, the intravenous OT infusion resulted in significantly higher AUC over the observation interval, while the intranasal methods did not differ in terms of AUC (Fig. 6b) (Supplementary Table 2) (Repeated measures One-way ANOVA: $F_{(1.15,24.24)} = 35.80$, $p < 0.001$, $\eta^2 p = 0.705$; post-hoc tests: Spray vs Nebulizer—$p = 0.980$; Spray vs Intravenous—$p < 0.001$; Nebulizer vs Intravenous—$p < 0.001$). There were no significant differences in the absolute bioavailability of OT absorbed to the plasma between the standard nasal spray and PARI SINUS nebulizer (Fig. 6c) (Supplementary Table 2) (Paired t-test: $T(15) = 0.796$, $p = 0.438$, d = 0.199).

**Regional temporal dynamics of pharmacological effects**. We present in Fig. 7 graphs depicting changes over time in regional perfusion in anatomical regions of interest (ROIs) corresponding to clusters showing significant OT-induced rCBF changes (compared to placebo) for each method of administration in the whole-brain univariate analyses. Overall, these graphs show that the temporal dynamics of the effects of synthetic oxytocin administration on brain perfusion is complex and shows regional specificities. Depending on the region of the brain, the effects can start or be maximal at different times post-dosing and extend over different periods of time. For instance, Fig. 7 shows that the OT-induced decrease in left amygdala perfusion is largely an early effect, maximal in the first few minutes (<25 min) after OT administration and independent of method of administration. In contrast, the OT-induced rCBF increase observed in the posterior

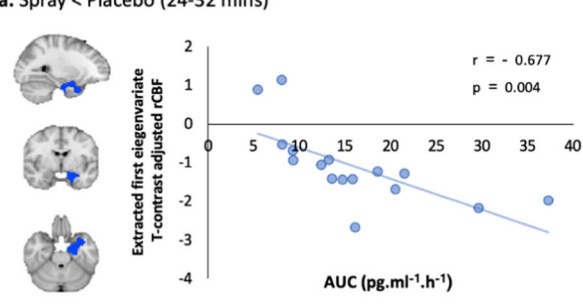

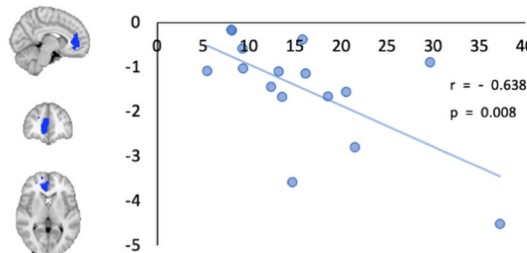

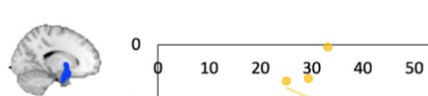

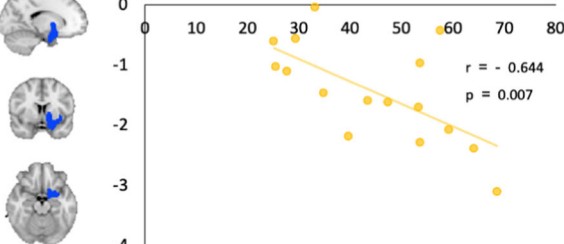

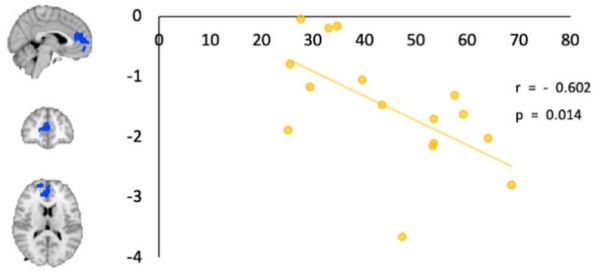

Fig. 5 Changes in rCBF associated with systemic OT are predicted plasma OT post-dosing. To investigate if concomitant increases in peripheral OT were related to treatment-induced changes in rCBF, we extracted data from each significant rCBF cluster in the paired sample t-tests and calculated Spearman correlation coefficients between these contrasts estimates and the areas under the curve (AUCs) reflecting individual differences in treatment-induced OT plasma concentrations for the corresponding method of administration. In panel **a**, we present decreases in rCBF after spray OT at 24–32 min post-dosing; in panel **b**, decreases in rCBF after spray OT at 87–95 min post-dosing; in panel **c**, decreases in rCBF after intravenous OT at 24–32 min post-dosing; and in panel **d**, decreases in rCBF after intravenous OT at 87–95 min post-dosing. r corresponds to the Spearman coefficient of correlation. Statistical significance was set to $p < 0.05$ (two-tailed). Source data are provided as a Source Data file.

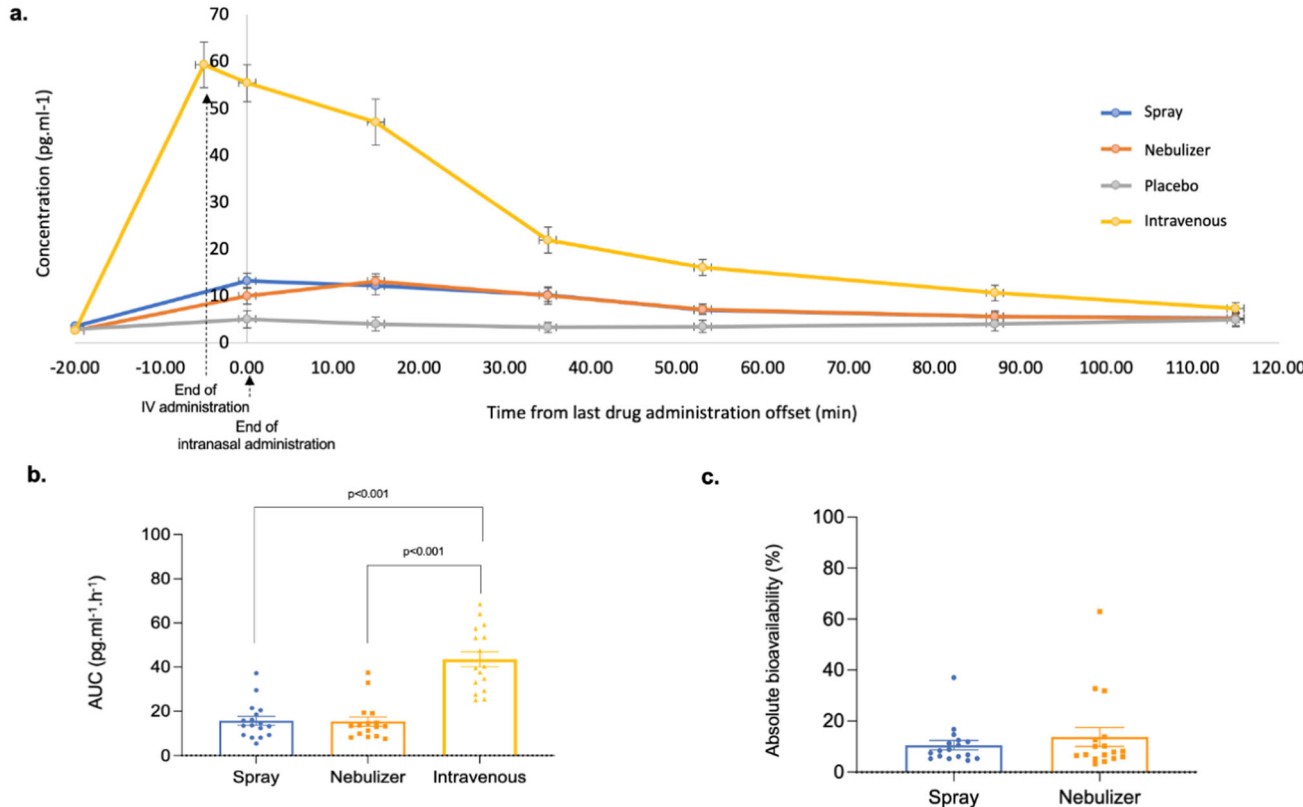

**Fig. 6 Spray and nebulized OT result in similar pharmacokinetics profiles in the plasma.** From the plasmatic concentrations of OT at several timepoints after dosing, we calculated the area under the curve (AUC) using the trapezoidal rule for each subject and session. We also determined the absolute bioavailability of OT in systemic circulation for each of the intranasal methods of administration, calculated as the dose-corrected AUC of each intranasal administration divided by the dose-corrected AUC of the intravenous administration. Mean AUCs were compared between the three methods of administration with a repeated measures one-way analysis of variance, using the Greenhouse-Geisser correction against violations of sphericity. Post-hoc comparisons between methods were implemented using Tukey's correction for multiple comparisons. Mean absolute bioavailabilities were compared between the standard nasal spray and the *PARI SINUS* nebulizer using a two-tailed paired *t*-test. In panel **a**, we present the variations of the concentrations of OT in plasma across time; in panel **b**, we present the results of the comparison of AUCs between treatments; and in panel **c**, we present the comparison of absolute plasmatic bioavailability between spray and nebulizer. Data are presented as mean ± 1 standard error of the mean (SEM). Error bars correspond to SEM. Statistical significance was set to $p < 0.05$ (two-tailed). Source data are provided as a Source Data file.

insula after administration by standard nasal spray tends to be more sustained in time and maximal at later time-intervals.

**Heart rate and heart-rate variability**. There were no significant main effects of treatment or time-interval, and no significant treatment × time-interval effects on heart rate or on any time domain, frequency domain or nonlinear measures of heart-rate variability (Supplementary Table 5).

## Discussion
This is the first in man study to investigate the pharmacodynamics of synthetic oxytocin on resting rCBF over an extended period of time when administered intravenously, with a nebulizer or a standard nasal spray. Our study yielded three key findings, which we discuss below in turn.

Our first key finding was the observation of OT-induced decreases in rCBF in the left amygdala and the anterior cingulate cortex with both the intravenous and standard nasal spray administration methods at overlapping temporal intervals. These decreases in rCBF in both the left amygdala and anterior cingulate cortex correlated with nasal spray or intravenous-induced increases in OT plasma concentrations, and became nonsignificant when these concomitant changes in OT plasma concentration were added as a covariate in the model. At the same

time, concomitant change in plasma OT concentration did not correlate with or account for any of the remaining changes in rCBF, when OT was administered intranasally either with a standard nasal spray or the nebulizer.

The suppression of amygdala's activity constitutes one of the most robust findings in animal studies and intranasal OT studies in men[34,37–39]. Similarly, human BOLD fMRI studies have implicated intranasal OT-induced decreases in BOLD in the anterior cingulate cortex in the modulation of social cognition[40], emotion[41] or fear consolidation[42] effects. These suppressive effects on BOLD match our observation of decreases in rCBF in these areas at rest—which we suggest is likely to reflect decreases in local metabolic demands associated with decreasing neural activity at rest. We provide first evidence that the intravenous infusion of OT echoes the effects of a standard spray administration on brain's physiology within key neural circuits at rest. These effects are consistent with previous observations of improved repetitive behaviours and social cognition in ASD patients after intravenous administration of OT[21–24] and provide a possible mechanism by which these therapeutical effects may arise. Therefore, our findings challenge the current assumption that key effects of intranasal OT on brain function and behaviour are entirely derived by direct nose-to-brain transport.

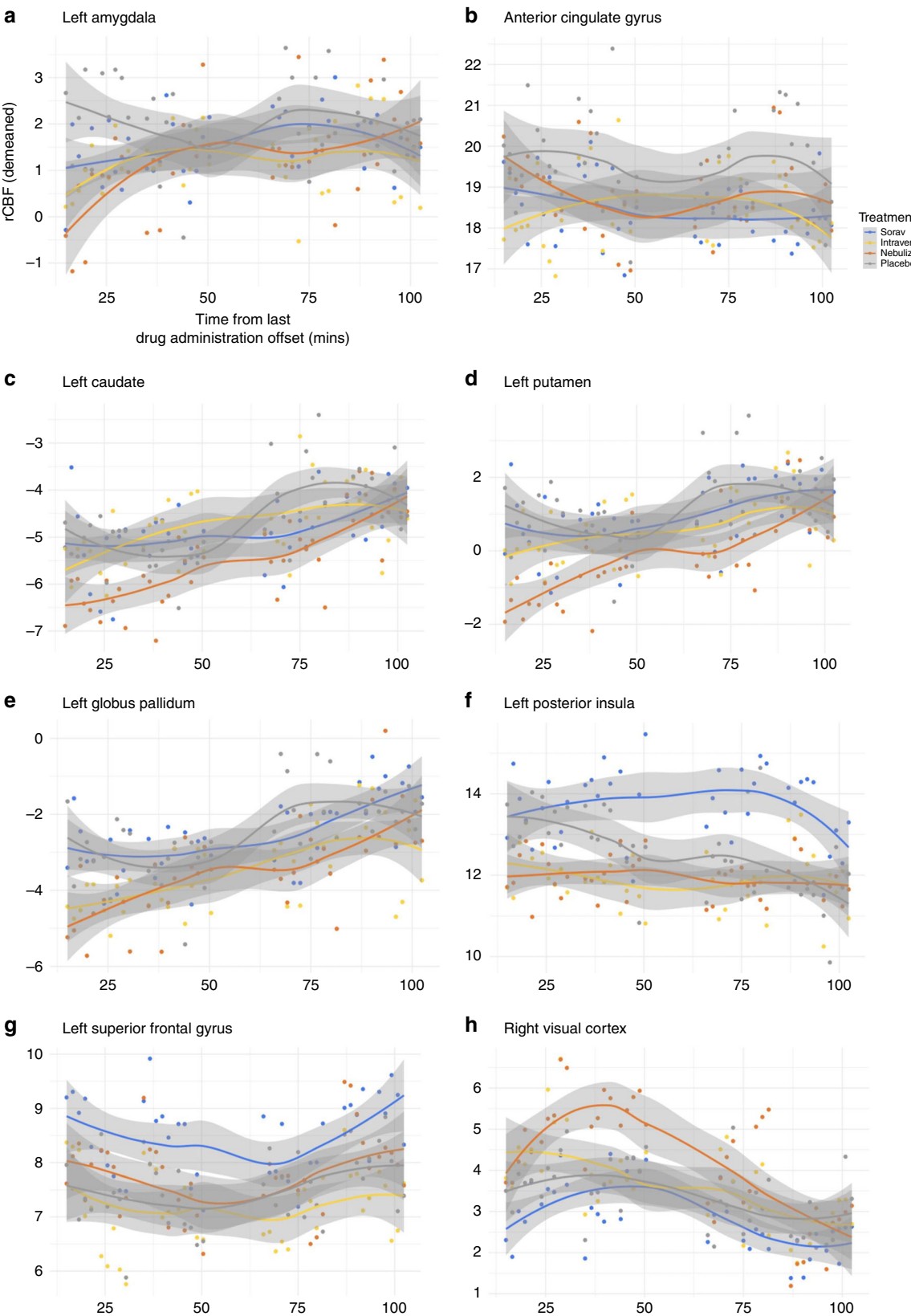

With respect to changes in rCBF induced by the intravenous administration of OT, there may be three possible mediating mechanisms. First, it is possible that peripheral effects of OT on vegetative territories, such as the heart, may be an indirect source of changes in areas of the interoceptive/allostasis network in the human brain, where the dorsal amygdala and the pregenual anterior cingulate cortex assume the role of visceromotor hubs[43]. However, as we did not observe any effects of OT (irrespective of

**Fig. 7 Regional temporal dynamics of rCBF changes in key anatomical regions of interest.** In order to illustrate the time-course of the changes in rCBF for each method of administration in key brain areas showing significant OT-induced changes in rCBF in the whole-brain univariate analyses, we estimated and plotted mean rCBF values over-time from anatomical regions-of-interest (ROIs) corresponding to the identified significant clusters. Each subplot depicts mean rCBF values (y-axis) at each time-interval (x axis), for each method of administration in each ROI. Regression lines were fitted using a non-parametric regression method that fits multiple regressions in local neighbourhood (Locally estimated scatterplot smoothing - LOESS) to the time-course of mean rCBF in each ROI for each method of administration separately (the shadows denote the 95% confidence interval for each fitted line)[97]. To avoid the potential confounding effects of nominal changes in global CBF across time or between methods of administration, we used demeaned values of mean ROI rCBF estimates, which correspond to subtracting global CBF estimates from the raw mean rCBF in each ROI. Source data are provided as a Source Data file.

administration method) on heart rate or heart-rate variability, this cannot explain the decreases in rCBF we observed in this study. Second, it is possible that the small amounts of OT that cross the BBB[15] (or a functional metabolite) is sufficient to induce changes in rCBF in brain regions of high density of the OT receptor, such as the amygdala, but not in other regions where the lower availability of the receptor would require higher local concentrations[44]. This hypothesis is in line with three recent studies. The first study reported that the intravenous infusion of labelled synthetic OT increased synthetic OT levels in the CSF in primates[45]. The second study found that synthetic OT when administered intraperitoneally increased the concentration of OT in the extracellular fluid in the amygdala and dorsal hippocampus, both in wild-type and OT knockout mice[46]. The third study showed that circulating OT can be transported into the brain[47]. Third, it is possible that the small amount of synthetic OT crossing the BBB is sufficient to engage OT-synthesising neurons in the hypothalamus, inducing the release of endogenous OT in the brain[48]. However, a recent study in primates that administered labelled OT did not support this hypothesis[45].

Our second key finding was the observation of increases in rCBF following intranasal administration (using either a standard nasal spray or the nebulizer), which could not be explained by concomitant increases in plasma OT. Indeed, there were no significant increases in rCBF (even at a lower threshold) when OT was administered intravenously. This finding is consistent with the contribution of nose-to-brain pathways for these effects of intranasal OT in humans. It matches well recent evidence showing that the intranasal administration of OT in mice increased the concentrations of OT in the extracellular fluid of the amygdala beyond those achieved with the intraperitoneal administration of a dose matched to result in similar plasmatic elevations[46].

Our third key finding was that while the application of the same nominal dose of intranasal OT (40IU) with the standard nasal spray and the nebulizer resulted in identical pharmacokinetic profiles, the patterns of OT-induced changes in rCBF were markedly different across the two methods of intranasal administration. Given the similarity in pharmacokinetic profiles, we hypothesize that the difference in the patterns of OT-induced rCBF changes achieved with each method can only be explained by differences in the deposition of OT in the olfactory and respiratory regions and the parasinusal cavities, which receive innervation from the olfactory and trigeminal nerves and may thus constitute points of entry to the brain. It is possible that, as expected, the nebulizer achieved higher OT deposition in these areas[32] and hence resulted in increased amounts of OT reaching the brain (compared to the standard nasal spray). Consistent with this hypothesis, we found that OT induced increases in rCBF in brain regions more posterior and distant from the point of entry, such as the visual cortices. These areas, although expressing relatively low levels of the OTR, are enriched in the expression of the vasopressin receptor 1 (V1aR) mRNA (Supplementary Fig. 4); OT has considerable affinity for the V1aR in higher

concentrations[49]. The higher central concentrations of OT achieved with the nebulizer may have allowed the peptide to diffuse farther and hence activate the V1aR that would not have been targeted by lower local concentrations of the peptide. This hypothesis assumes that the amounts delivered with the nasal spray do not saturate the nose-to-brain pathway, if it exists.

The fact that the nebulizer resulted in a different pattern of rCBF changes, with null or minimal overlap with the changes observed after the standard spray, instead of simply observing changes in the magnitude of the effects within the same areas, is surprising to some extent. We hypothesized that the nebulizer would result in changes similar to those achieved by the spray but of higher magnitude. We also considered that the nebulizer might modulate rCBF in areas that were not targeted either by the spray or the intravenous administrations. However, this prediction would only have been valid if the pharmacodynamics of the rCBF response to OT followed a linear model—which does not seem to be the case at least for some brain areas. The few studies that have inspected the dose-response effects of intranasal OT on the BOLD response in the amygdala support an inverted-U shape curve of response by showing that deviating from an "optimal" dose may in fact result in lower or null effects[35,50].

We believe the complexity of the central OT signalling machinery[51] should be considered to interpret these findings. The OTR has been described to recruit different intracellular G protein ($G_s$ or $G_i$) pathways, depending on ligand, receptor and G protein type distribution and abundance[44,51]. $G_s$ and $G_i$ activation typically result in opposite effects in terms of cellular function[52], meaning that in areas of high density of $G_i$ proteins higher amounts of OT may in fact result in inhibition of neuronal activity or null effects[44,51]. This complexity might explain, for instance, why we did not observe changes in rCBF with the nebulizer in regions where the standard nasal spray produced effects. Additionally, the lack of OT-induced rCBF changes, when OT was administered with the nebulizer, in some of the clusters where we noted significant effects for the nasal spray (such as the left amygdala and the ACG), might simply be a result of statistical thresholding. For example, plots of OT-induced changes in rCBF in the left amygdala, over time, (see Fig. 6) revealed similar OT-induced decreases in rCBF across all three administration methods at the first two time intervals. The lack of a significant OT-induced decrease following administration with the nebulizer in the whole-brain univariate analysis is likely to reflect the higher variability of decreases in rCBF, across participants, compared to the other methods, at these time intervals. Until a ligand allowing for direct quantification of in vivo penetration of OT in the brain after intranasal administration might be produced, a dose-response study using the nebulizer may allow us to gain further indirect insights about whether using the nebulizer may confer certain advantages regarding targeting the central OT system.

While our findings are consistent with the idea that nose-to-brain pathways could contribute to some of the changes in rCBF induced by intranasal OT, our study cannot illuminate the precise mechanisms underlying these effects. We believe that most

evidence to date concurs on the idea that OT, when administered intranasally, may diffuse from the olfactory and respiratory epithelia in the middle and upper posterior regions of the nasal cavity along ensheathed channels surrounding the olfactory and trigeminal nerve fibre pathways and lymphatic vessels to the CSF and/or the brain[53]. Research in rodents and monkeys has shown that molecules administered intranasally can be transported to the olfactory bulb within a time-frame of 45–90 min[54,55], and possibly much faster[56–58]. A recent comparison of intranasal and intravenous administration of a new OT receptor tracer in mice supported this hypothesis by showing uptake of the intranasally administered tracer to the olfactory bulb, while increases in this area after the intravenous administration of the tracer were almost negligible[59]. The increases of the concentration of the tracer in the olfactory bulb following its intranasal administration could be observed as soon as 30 min post-dosing, which fits the time-frame of the effects we report in our study. While our findings cannot illuminate the precise pathway through which intranasal OT may reach the brain, the fact that we observe distinct patterns of changes in rCBF with two different intranasal methods suggests that the changes we see in the brain are unlikely to be explained by local effects of intranasal OT in the nasal cavity. Expression of the OT receptor has been reported in human taste buds[60] and in the rat olfactory epithelium[61]. Direct actions of OT on olfactory and trigeminal nerve bundles could parsimoniously explain the discrepancies between the changes observed after intranasal and intravenous administration of OT. If direct modulation of activity in these nerve bundles would account for all the changes we report for our intranasal administration methods, then one would expect that the changes observed in rCBF after intranasal administration of synthetic OT would be mostly restricted to the olfactory/trigeminal pathways and respective connected areas, which is not entirely supported by our data. In fact, the effects we observed after synthetic OT map to areas where expression of receptors for OT (either the OTR or the AVPR1A) seem to be present, as per our current understanding of the distribution of these receptors in the human brain[62,63]. While we cannot exclude that some of the effects we observed after intranasal OT may relate to direct modulation of activity in nerves bundles, this mechanism is unlikely to fully account for all the effects we report herein.

Finally, we note that assuming a unique pharmacodynamics model for the effects of synthetic OT on rCBF, with a single time-course, may be simplistic. In fact, our data suggest that depending on the region of the brain, changes in rCBF, compared to placebo, can start or be maximal at different times post-dosing and extend over different periods of time. This regional diversity in pharmacodynamics is not surprising given the known complexity of the OT signalling machinery, regional variations in receptors density and in the distribution of OT in the brain as a function of entry point[44,51]. Our study may help to guide future studies aiming to target specific regions by providing an indication of the time-intervals post-dosing at which these effects may be maximal.

From a translational perspective, our findings emphasize the inadequacy of a one-fits-all approach in the administration of synthetic OT to target the central OT system in humans for the treatment of brain's disorders. Our findings indicate that some specificity may be achieved depending on the route used to deliver OT. Given that enhancement of brain's metabolism in areas such as the frontal gyrus, insula or occipital cortices may be restricted to the intranasal route, clinical applications aiming to target these circuits should thus prefer this route. However, we should not completely discard the potential utility of the peripheral route. Systems of controlled sustained drug release for the peripheral circulation (i.e. transdermal controlled release[64]) already in place may provide an excellent opportunity to explore the clinical value of this route during chronic administrations in patients.

Our study faces certain limitations. First, the amount of OT administered intravenously was not chosen to mimic exactly the plasmatic concentrations achieved after intranasal administration[50]. Instead, we adopted a proof-of-concept approach, aiming to achieve consistently higher plasmatic concentrations of OT during the full period of scanning. Future studies should include an intravenous comparator that achieves pharmacokinetic profiles that are similar to those achieved with the intranasal methods. Second, our findings cannot be readily extrapolated to women, given the known sexual dimorphism of the OT system[65–67]. Third, although we tested for the potential effects of synthetic OT on cardiac physiology as a confounder, we acknowledge that potential effects on other peripheral systems need consideration. Future studies need to compare the intranasal and intravenous administration of OT with the parallel administration of a specific non-brain penetrant OT receptor antagonist to isolate the potential contribution of OT's signalling in the periphery. Fourth, we relied on a biomarker of brain physiology (rCBF) to probe the effects of different methods of administration of OT on brain function. This method is an indirect but sensitive signature of regional neuronal activity[68,69]; however, it is theoretically susceptible to potential confounding vascular effects[70]. While we cannot exclude the presence of such effects, we do not believe they are an important confounder of our results for two reasons: first, if the effects we observed after synthetic OT solely reflected drug effects on vasculature, then we should expect to see the same effects when administering OT intranasally and intravenously as both routes increase the concentration of the peptide in the systemic circulation to supraphysiologic levels—this is not what we found; second, vascular effects are likely to be reflected on changes in global brain perfusion, but our results did not show any significant effects of any treatment on global CBF—in fact, we included global CBF as a nuisance variable in all of our analyses, therefore accounting for unspecific global vascular effects on CBF. The idea that the reported effects of OT on rCBF do not reflect unspecific vascular effects on global perfusion is further supported by preclinical data showing that the modulation of rCBV by intranasal OT in rodents is likely to result from neural activity[71]. Finally, the univariate analyses we present in this paper should not be used to define the temporal dynamics of the effects of synthetic OT on rCBF (as has been previously explored using pattern recognition analyses)[16] because of the risk of false negatives in specific time-intervals. Instead, we provide the time-courses of perfusion in the areas where we found significant OT-induced changes in rCBF in the whole-brain univariate analyses (Fig. 7), which the reader may use to gain insight about the diversity of the pharmacological effects on a regional basis.

In conclusion, we provide evidence consistent with the involvement of nose-to-brain pathways to the changes in brain physiology observed after intranasal synthetic OT in humans. At the same time, we also demonstrate that some of the key effects of synthetic OT in the human brain, when delivered by standard nasal sprays, can be explained by concomitant increases in peripheral OT levels post-dosing. Our results emphasize the inadequacy of a one-fits-all approach in the administration of synthetic OT to modulate brain function for the treatment of psychiatric or neurological conditions in humans, while highlighting the importance of optimizing the delivery of peptides to the brain through nose-to-brain pathways.

## Methods

**Participants**. We recruited 17 healthy male adult volunteers (mean age 24.5, SD = 5, range 19–34 years). One participant did not complete one of the four visits and for this reason was excluded from all analyses. We screened participants for

psychiatric conditions using the Symptom Checklist-90-Revised[72] and the Beck Depression Inventory-II[73] questionnaires. Participants were not taking any pre-scribed drugs, did not have a history of drug abuse and tested negative on a urine panel screening test for recreational drugs, consumed <28 units of alcohol per week and <5 cigarettes per day. We instructed participants to abstain from alcohol and heavy exercise for 24 h and from any beverage or food for 2 h before scanning. Participants gave written informed consent. King's College London Research Ethics Committee (PNM/13/14-163) approved the study. We determined sample size based on our previous validation study demonstrating that $N = 16$ per group was sufficient to quantify standard nasal spray OT-induced changes in rCBF in a between-subjects design[16,74].

**Study design.** We employed a double-blind, placebo-controlled, triple-dummy, crossover design. Participants visited our centre for four experimental sessions spaced 8.90 days apart on average (SD = 5.65, range: 5–28 days). In each session, participants received treatment via all three administration routes, in one of two fixed sequences: either nebulizer/intravenous infusion/standard nasal spray, or standard nasal spray/intravenous infusion/nebulizer, according to the treatment administration scheme presented in Fig. 1. In three out of four sessions only one route of administration contained the active drug; in the fourth session, all routes delivered placebo or saline. Participants were randomly allocated to a treatment order (i.e. a specific plan regarding which route delivered the active drug in each experimental session) that was determined using a Latin square design. Unbe-known to the participants, the first treatment administration method in each session always contained placebo (see Administration 1 in Fig. 1), while intranasal (spray or nebulizer) OT was only delivered with the third treatment administra-tion. This protocol maintained double-blinding while avoiding the potential washing-out of intranasally deposited OT (as might be the case if OT had been administered at the first treatment administration point and placebo at the third administration point).

**Intranasal OT administration.** For the intranasal administrations, participants self-administered a nominal dose of 40 IU OT (Syntocinon; 40IU/ml; Novartis, Basel, Switzerland), one of the highest clinically applicable doses[75]. We have shown that 40IU delivered with a standard nasal spray induce robust rCBF changes in the human brain in a between-subjects, single-blind design study[16]. For the intranasal administration, we used specially manufactured placebo that contained the same excipients as Syntocinon except for oxytocin.

*Standard nasal spray administration:* Participants self-administered 10 puffs, each containing 0.1 ml Syntocinon (4IU) or placebo, one puff every 30 s, alternating between nostrils (hence 40IU OT in total). The aerosol droplet size of three Syntocinon spray bottles was assessed by laser diffraction (Malvern Spraytec, Malvern Panalytical, Worcestershire, UK). The mass median diameter (MMD) of the aerosol plume was $37 \pm 2.5$ μm.

*PARI SINUS nebulizer:* Participants self-administered 40IU OT (Syntocinon) or placebo, by operating the SINUS nebulizer for 3 min in each nostril (6 min in total), according to instructions. The correct application of the device was confirmed by determining gravimetrically the administered volume. Participants were instructed to breathe using only their mouth and to keep a constant breath rate with their soft palate closed, to minimize delivery to the lungs. The *PARI SINUS* (PARI GmbH, Starnberg, Germany) is designed to deliver aerosolised drugs to the sinus cavities by ventilating the sinuses via pressure fluctuations. The SINUS nebulizer produces an aerosol with 3 μm MMD, which is superimposed with a 44 Hz pulsation frequency. Hence, droplet diameter is roughly one tenth of a nasal spray and its mass is only a thousandth. The efficacy of this system was first shown in a scintigraphy study by ref. [76]. Since the entrance of the sinuses is located near the olfactory region, an improved delivery to the olfactory region was expected compared to nasal sprays. Other studies[32] have shown up to 9.0% (±1.9%) of the total administered dose to be delivered to the olfactory region, 15.7 (±2.4%) to the upper nose.

**Intravenous OT administration.** For the intravenous administration, we delivered 10IU OT (Syntocinon injection formulation, 10IU/ml, Alliance, UK) or saline via slow infusion over 10 min (1IU/min). A 50-ml syringe was loaded with either 32 ml of 0.9% sodium chloride (placebo) or 30 ml of 0.9% sodium chloride with 2 ml of Syntocinon (10IU/ml). A Graseby pump was used to administer 16 ml of the compound (hence 10IU of OT in total) over 10 min, at a rate of 96 ml/h. The ECG was monitored during the intravenous administration interval. We selected the intravenous dose and rate of administration to assure high plasmatic concentra-tions of OT throughout the observation period while restricting cardiovascular effects to tolerable and safe limits. A rate of 1IU per minute is typically used in caesarean sections and is considered to have minimised side effects[77,78].

**Procedure.** Each experimental session began with the treatment administration protocol that lasted about 22 min in total (Fig. 1). After drug administration, participants were guided to the MRI scanner, where eight pulsed continuous arterial spin labelling scans (each lasting approx. eight minutes) where acquired, spanning 15–104 min post-dosing, as detailed in Fig. 1. Participants were instructed to lie still and maintain their gaze on a centrally placed fixation cross during

scanning. We assessed participants' levels of alertness (anchors: alert-drowsy) and excitement (anchors: excited-calm) using visual analogue scales (0–100) at three different timepoints during the scanning session (the first one immediately before the first scan—around 15 mins post-dosing, the second one immediately after the fourth scan—around 55 mins post-dosing and the last one immediately before the seventh scan—around 92 mins post-dosing) to evaluate subjective drug effects across time. An 8-min resting state BOLD fMRI scan was obtained at about 60 min post-dosing (data not presented here).

**Blood sampling and plasmatic OT quantification.** We collected plasma samples at baseline and at five timepoints post-dosing (as detailed in Fig. 1) to measure changes in the concentration of OT. Plasmatic OT was assayed by radio-immunoassay (RIAgnosis, Munich, Germany) after extraction, currently the gold-standard technique for OT quantifications in peripheral fluids[79]. The detection limit of this assay is in the 0.5 pg/sample range. Cross-reactivity with vasopressin, ring moieties and terminal tripeptides of both OT and vasopressin and a wide variety of peptides comprising 3 (alpha-melanocyte-stimulating hormone) up to 41 (corticotrophin-releasing factor) amino acids are <0.7% throughout. The intra- and inter-assay variabilities are <10%[80].

**MRI data acquisition.** We used a 3D pseudo-continuous Arterial Spin Labelling (3D-pCASL) sequence to measure changes in regional Cerebral Blood Flow (rCBF) over 15–120 min post-dosing. Labelling of arterial blood was achieved with a 1525 ms train of Hanning shaped RF pulses in the presence of a net magnetic field gradient along the flow direction (the *z*-axis of the magnet). After a post-labelling delay of 2025 ms, a whole-brain volume was read using a 3D interleaved "stack-of-spirals" Fast Spin Echo readout[81], consisting of eight interleaved spiral arms in the in-plane direction, with 512 points per spiral interleave. TE was 11.088 ms and TR was 5135 ms. 56 slice-partitions of 3 mm thickness were defined in the 3D readout. The in-plane FOV was 240 × 240 mm. The spiral sampling of k-space was re-gridded to a rectangular matrix with an approximate in-plane resolution of 3.6 mm. The sequence acquired 5 control–label pairs. Individual CBF maps were computed for each of the perfusion weighted difference images derived from every control–label (C–L) pair, by scaling the difference images against a proton density image acquired at the end of the sequence, using identical readout parameters. This computation was done according to the formula suggested in the recent ASL consensus article[82]. The sequence uses four background suppression pulses to minimise static tissue signal at the time of image acquisition. We performed eight of these 3D-pCASL sequence acquisitions, the acquisition time of each sequence was 8:20 min. A 3D high-spatial-resolution, Magnetisation Prepared Rapid Acquisition (3D MPRAGE) T1-weighted scan was acquired. Field of view was 270 mm, TR/TE/TI = 7.328/3.024/400 ms. The final resolution of the T1-weighted image was $1.1 \times 1.1 \times 1.2$ mm.

**MRI data preprocessing.** A multi-step approach was performed for the spatial normalization of the CBF maps computed for each C–L pair to the space of the Montreal Neurological Institute (MNI): (1) co-registration of the proton density image from each sequence to the participant's T1-image after resetting the origin of both images to the anterior commissure. The transformation matrix of this co-registration step was then applied to the CBF map from each C–L pair, to trans-form the CBF map to the space of the T1-image; (2) unified segmentation of the T1 image; (3) elimination of extra-cerebral signal from the CBF map, by multiplication of the "brain only" binary mask obtained in step [2], with each co-registered CBF map; (4) normalization of the subject's T1 image and the skull-stripped CBF maps to the MNI152 space using the normalisation parameters obtained in step [2]. Finally, we spatially smoothed each normalized CBF map using an 8-mm Gaussian smoothing kernel. All of these steps were implemented using the ASAP (Automatic Software for ASL processing) toolbox (version 2.0)[83]. The resulting smoothed CBF maps from each C-L pair were then averaged, using the fslmaths command implemented in the FMRIB Software Library (FSL) software applications (https://fsl.fmrib.ox.ac.uk/fsl/fslwiki/), to obtain a single averaged CBF map for each of the time-intervals depicted in Fig. 1.

**Physiological data acquisition and processing.** Heart rate was continuously monitored during the scanning period using MRI-compatible finger pulse oximetry while the participant rested in supine position, breathing spontaneously in the scanner. The data were recorded digitally as physiologic waveforms at a sampling rate of 50 Hz. Heart beats were firstly automatically detected using an in-house script and then visually inspected and manually cleaned for misidentified beats. Inter-beat interval values were then calculated. The resulting cleaned data were then transferred to *Kubios HRV analysis software* (MATLAB, version 2 beta, Kuopio, Finland) and a set of time domain (heart-rate (HR) and the root mean square of the successive differences (RMSSD), frequency domain (low (LF) and high (HF) frequencies spectral powers and high/low frequency spectral power ratio (HF/LF)) and nonlinear (Approximate entropy (ApEn), the SD1 and SD2 lines from the Poincare Plot and the detrended fluctuation scaling exponents DFAα1 and DFAα2) analysis measures were calculated using the default *Kubios* pipeline[84,85]. We decided to examine a wide-range of different heart variability measures because previous studies have diverged in the metrics where they found

effects of OT on heart-rate variability[84]. For instance, there is currently debate whether time and frequency analysis measures can be sufficiently sensitive to capture important (nonlinear) changes in heart-rate time series, including those changes associated with OT administration[85–87]. In addition to the manual cleaning of the data, we also employed a threshold-based method of artefact correction, as provided by *Kubios*, where artefacts and ectopic beats were simply corrected by comparing every RR interval value against a local average interval. The threshold value used was 0.35 sec. Following current recommendations for heart-rate data processing and analysis, if more than 5% of the beats required correction, then we decided to exclude these periods of observation[88]. All of these metrics were calculated based on the data recorded for the whole duration of each scan (8 min). In some rare cases where artefacts could not be corrected, we only included the data if at least 5 min of acquisition free of artefacts could be analysed. We based our decision on the fact that at least 5 min of observation are required for pulse plethysmography to reflect heart-rate variability as assessed by electrocardiography[89]. The percentages of the total amount of available data used for this analysis after data quality control can be found in Supplementary Table 4.

**Statistical analyses**. *Global CBF Measures*: We extracted mean global CBF values within an explicit binary mask for grey-matter (derived from a standard T1-based probabilistic map of grey-matter distribution by thresholding all voxels with a probability >0.20) using the fslmeants command implemented in the FSL software suite. We tested for the main effects of treatment and time-interval and for the interaction between both factors on global CBF signal in a repeated measures analysis of variance Treatment and Time as factors, implemented in SPSS 24 (http://www-01.ibm.com/software/uk/analytics/spss/), using the Greenhouse-Geisser correction against violations of sphericity.

*Subjective ratings*: For the two subjective ratings of alertness and excitement collected at the three timepoints post-dosing, we initially tested for the main effects of treatment and time interval and for the interaction between both factors, as previously described for global CBF. Second, we investigated the association between changes in global CBF signal and self-ratings of alertness and excitement over time, using within-group pooled correlation coefficients. For equal variances of the correlated variables in the four subgroups, pooled within-group correlation coefficients represent a weighted mean of the within-group correlation coefficients, weighted by the number of observations in each subgroup[90]. They correspond to the result of statistically eliminating subgroup differences from the total group correlation coefficient. Since we only collected ratings at three timepoints, for this analysis we selected the global values of the scans that were closer in time to the moment of the ratings acquisition. We firstly inspected the equality of the covariance matrixes for each rating scale across the four treatment groups to decide whether to pool the four groups or not, using the *mconvert* command at SPSS. We then calculated the association between each rating scale and global CBF for each participant and averaged the covariance matrices to estimate the pooled within-group Pearson correlation coefficients. Results are reported at a level of significance α = 0.05.

*Whole-Brain univariate analyses*: We firstly implemented an analysis of covariance design, controlling for global effects on CBF, using a flexible factorial model in SPM12 software (http://www.fil.ion.ucl.ac.uk/spm/software/spm12/) where we specified the factors Subjects, Treatment, and Time interval. We used an F test to investigate the main effect of treatment, i.e. identify brain regions where any treatment induced persistent changes in rCBF (regardless of direction) across time and the main effect of time interval, i.e. identify brain regions where any time-interval induced changes in rCBF (regardless of direction), despite treatment. We also used an F test to investigate the interaction of Treatment × Time-interval i.e. identify brain regions showing specific treatment-induced changes in rCBF as a function of time interval, regardless of direction. Our study is novel in multiple ways: it is the first study to investigate the pharmacodynamics effects of synthetic oxytocin on resting brain physiology over an extended period of time when administered with any of the three methods of administration we used in a double-blind placebo-controlled crossover design. It is also the first in man study regarding the effects of OT administered intravenously or with a nebulizer on rCBF. Since our study maps uncharted territory, we followed up the flexible factorial model with an exhaustive series of paired *t*-tests at each time interval to investigate the direction of potential OT-induced changes in rCBF specifically for each treatment route (compared to placebo). We conducted whole-brain cluster-level inference for all analyses, reporting clusters significant at α = 0.05 using familywise error (FWE) correction and a cluster-forming threshold of $P = 0.005$ (uncorrected). Our statistical thresholds were determined a priori based on our own previous work investigating the effects of intranasal spray OT on rCBF in humans[16] and are standardly applied in pharmacological ASL studies measuring rCBF[91–96]. For the paired *t*-tests, in recognition of the increased risk of false positives given the large number of paired *t*-tests ($3 \times 8 = 24$), we mark clusters that do not survive correction for multiple testing following adjustment of the P-value using conservative Bonferroni correction for the 24 paired-*t*-tests performed with an asterisk (*) ($P_{adjusted} = 0.05/24 = 0.002$) (see Tables 2–4). Given the well-known decrease in global CBF across time (and in the absence of a treatment × time interval interaction on global CBF values—see Results)[16], we included global CBF values as a nuisance covariate in our general linear model to

enhance sensitivity to detect OT-induced changes in regional CBF/neuronal activation.

*Regional temporal dynamics of pharmacological effects*: We explored the time-course of rCBF changes by estimating the mean rCBF values using the fslmeants command in anatomical regions of interest (ROIs) corresponding to clusters showing significant OT-induced rCBF changes (compared to placebo) for each method of administration in the whole-brain univariate analyses. Specifically, we defined anatomical ROIs for the left amygdala, anterior cingulate gyrus, left caudate, left putamen, left globus pallidum, right visual cortex, left posterior insula and left superior frontal gyrus using the Harvard-Oxford cortical and subcortical atlases distributed with FSL. To maximize the amount of data points used to fit the time-course functions (from 8 to 40 timepoints), at a cost of including noisier estimates, we estimated mean ROI rCBF values of each of the 5 rCBF maps corresponding to the 5 C-L pairs that were averaged to obtain the rCBF map for each of the 8 time-intervals. We estimated the rCBF time-course for each ROI implementing a non-parametric regression method that fits multiple regressions in local neighbourhood (*Locally estimated scatterplot smoothing - LOESS*) using the ggplot2 package from R[97]. This approach allows us to capture possible nonlinear relationships between rCBF and time, while reducing the noise and without making any assumptions about the relationship between these two variables. To control for the potential confounding effects of the small changes in global CBF across time or between methods of administration, we used demeaned rCBF estimates, which correspond to estimating raw ROI mean rCBF minus global CBF values. Since we performed these informative analyses a posteriori following the advice of one of our reviewers, we did not run any statistical analysis on these extracted data, to avoid double dipping. Instead, we provide these time-courses to illustrate the temporal dynamics of OT-induced rCBF changes, by method of administration, for each key brain region showing significant OT-induced changes in rCBF in the univariate whole-brain analyses.

*Pharmacokinetic analysis*: From the plasmatic concentrations of OT at several timepoints after dosing we calculated the area under the curve (AUC) using the trapezoidal rule for each subject and session. AUC provides a single metric that reflects variations in plasma levels of OT concentration following OT/placebo administration for each session and each participant. We also determined the absolute bioavailability of OT in systemic circulation for each of the intranasal methods of administration, calculated as the dose-corrected AUC of each intranasal administration divided by the dose-corrected AUC of the intravenous administration. Mean AUCs were compared between the three methods of administration with a repeated measures one-way analysis of variance, using the Greenhouse-Geisser correction against violations of sphericity. Post-hoc comparisons between methods were implemented using Tukey's correction for multiple comparisons. Mean absolute bioavailabilities were compared between the standard nasal spray and the *PARI SINUS* nebulizer using a two-tailed paired *t*-test.

*Association between OT-induced changes in rCBF and plasma OT*: To investigate if concomitant increases in peripheral OT were related to treatment-induced changes in rCBF, we extracted data from each significant rCBF cluster (adjusting for each treatment comparison contrast) in the paired sample *t*-tests and calculated Spearman correlation coefficients between these contrasts estimates and the AUCs reflecting individual differences in treatment-induced OT plasma concentrations for the corresponding method of administration. In this specific case, we employed the Spearman correlation coefficient because the number of observations used to estimate the correlation is small, which does not allow for an accurate verification of all implicit parametric analysis assumptions.

*OT effects on cardiac physiology*: Previous human studies have shown the ability of OT to affect cardiac physiology. Specifically, these studies have suggested that intranasal OT increases HRV at rest[85]. HRV is an important index for the heart-brain interaction[98]. Changes in HRV are accompanied by changes in the activity of several areas of the brain, including the amygdala (one of the areas of the brain most commonly implicated in OT effects on brain function and behaviour)[99,100]. Thus, OT-induced changes in HRV, if existent, could account for, at least, some of the OT-induced changes in rCBF we identify herein. We compared mean HR, RMSSD, HF, LF, HF/LF ratio, ApEN, SD1, SD2, DFAα1 and DFAα2 between methods of administration to examine the extent to which OT administration induced changes in HR or heart-rate variability, by using a repeated measures two-way analysis of variance. We used treatment and time interval as factors. We determined main effects of time interval and treatment, as well as their interaction, and used the Greenhouse-Geisser correction against violations of sphericity. We contained the false-discovery rate (FDR) at α = 0.05 using the Benjamini-Hochberg procedure, which is a more powerful version of the Bonferroni adjustment that allows non-independence between statistical tests[101]. Original *p*-values (two-tailed) are reported alongside with values obtained after accounting for FDR.

All the analyses were conducted with the researcher unblinded regarding treatment condition[102]. Since we used a priori and commonly accepted statistical thresholds and report all observed results at these thresholds—the risk of bias in our analyses is therefore minimal, if not null.

**Reporting summary**. Further information on research design is available in the Nature Research Reporting Summary linked to this article.

## Data availability

Data can be accessed from the corresponding author upon reasonable request. A reporting summary for this Article is available as a Supplementary Information file. The source data underlying Figs. 5, 6 and 7, Supplementary Figs. 1 and 2, and Supplementary Table 5 are provided as a Source data file.

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

## Acknowledgements

We would like to thank Mr. Robert Taylor for his help in organizing the pulse ple-tismography data and Dr. Elena Makovac for her help with heart-rate variability data analysis. This study was part-funded by: an Economic and Social Research Council Grant (ES/K009400/1) to Y.P.; scanning time support by the National Institute for Health Research (NIHR) Biomedical Research Centre at South London and Maudsley NHS Foundation Trust and King's College London to Y.P.; an unrestricted research grant by PARI GmbH to Y.P.

## Author contributions

Y.P. designed the study; Y.P., S.V., J.L. collected the data; N.M., A.O. and S.M. provided medical supervision and carried out medical procedures; D.A.M. analyzed the data; U.S.

and S.W. provided new analytical tools; D.A.M. and Y.P. interpreted the results and wrote the first draft of the paper; M.M., M.H., F.Z., S.C.R.W., D.M., G.M. and A.F. revised the manuscript for intellectual content.

## Competing interests

The authors declare no competing interests. This manuscript represents independent research. The views expressed are those of the authors and not necessarily those of the NHS, the NIHR, the Department of Health and Social Care, or PARI GmbH.
