## [Peer Review File · Nature Communications]

Reviewers' Comments:

Reviewer #1:

Remarks to the Author:

The attached report is of interest to the community of oxytocin researchers who need to understand the mechanisms of action of the nasal and intravenous administration. It addresses an important issue and the paper is very well written.

I think the claims of this paper are overstated and it is not until one reads the end of the discussion that the authors acknowledge the limitations. This is then followed by a conclusion paragraph that returns to overstatements. They should be acknowledged much earlier and clearly in the abstract.

My main concern is the use of blood flow as an indicator of some action. It seems optimistic to infer OT action from this mechanism. Blood flow is linked to many parameters, also comprising vasopressin activation (i.e. you squeeze the vessels, you reduce the flow). Most importantly, the authors acknowledge that they do not know the exact mechanism of the reduction of blood flow, in particular if it is periphery-mediated or centrally-mediated. In lack of this, I think it is way premature to conclude that they have demonstrated that there is a nose-to-brain pathway.

it looks to me that the nebulizer deposits OT faster in olfactory epithelium? Is that correct? It is hard to tell from the images.

It is important to acknowledge a number of studies that have previously shown intravenous administration also improves social cognition (see work of Hollander)

It is a little unclear why the authors suggest that heart rate variability measures as the main assessment for peripheral response. There are many peripheral sites in the human body that could trigger brain release of oxytocin so the absence of an effect of OT on HRV cannot rule out other peripheral actions.

Reviewer #2:

Remarks to the Author:

In the present work, Martins et al. explore whether different routes of oxytocin (OT) administration lead to different effects on brain activity, using arterial spin labelling as a readout. This is an important issue as a large number of studies has investigated the effects of exogenous OT on the brain, but the biological mechanisms underlying IN OT action are still unclear. However, there are two critical problems with the current manuscript, one is methodological, and the second one is interpretational.

1) The overall design and analysis are rather weak because authors choose a cluster forming threshold of $p < 0.005$ uncorrected and then analyzed the interaction time x treatment with post hoc paired t tests for each treatment condition and the interval. This means that at least 30 paired t test maps were computed (3 treatments x 10 time intervals). This high number of tests, at this permissive threshold ($p < 0.005$ uncorrected) and with 16 subjects is very likely to produce false positive results. This could explain the incoherence in the results (please see 2)).

There are several other minor issues with the methods and results:

- Clusters in which an interaction effect was seen (Fig3) do not match with the clusters displaying a main effect of time (Fig 2).
- Time intervals between figures, text and tables do not match (24-32min in fig3 and text, but 19-26min was used in the table?)

2) The results are not coherent. Firstly, the authors do not reproduce their previous findings: in their previous work (Paloyelis et al. 2016), the authors found that "the temporal profile of IN-OT-induced rCBF changes showed a peak response 39–51 min after IN-OT, followed by a gradual diminution of effects." However, in the present article, they fail to see any effect of IN OT (nor IV and nebulizer) during this precise time period (but they do before and after, which is even more confusing). This critical issue is not mentioned in the Discussion. How could oxytocin exert effects, then it stops for 30 min and then produces other effects? Furthermore, most studies employing IN OT in humans have precisely used this 45-75 min post administration interval to test their subjects and to find modifications of brain activity, which contradicts the absence of effects at this time interval in the present study. Secondly, it is hardly conceivable that the nebulizer produces effects completely different than the spray or intravenous administration. The nebulizer optimizes the amount of OT delivered to the posterior regions of the nasal cavity, but it does not permit OT to access new regions. Thus, it should lead to stronger effects than IN OT and, at the very least, we should see some overlap between the nebulizer and the spray effects. This incoherence is present throughout the discussion: authors first state that some of the IN OT effects are mediated by the increase of plasma OT, while some other effects (increase in rCBF) are mediated by a direct nose to brain pathway. This indicates, again, that IN OT effects should be in accordance with the ones obtain after nebulizer administration. The authors' explanation that OTR binds to different G protein is extremely far stretched, as the distribution of OTR is still very unclear in the human brain, so inferring regional differences in G protein binding is purely theoretical, and does not explain a complete mismatch between regions modulated by the spray and the nebulizer.

Reviewer #3:

Remarks to the Author:

This paper "Do direct nose-to-brain pathways underlie intranasal oxytocin-induced changes in regional cerebral blood flow in humans?" bravely enters a veritable minefield.

In recent years there has been a massive proliferation of interest in the brain oxytocin system and its role in not only social behavior but also appetite regulation and pain processing. Much of that interest has been engaged by a now vast number of studies in humans using intranasal application of oxytocin. However despite the volume of published work there remains at best a minimal understanding of how much oxytocin actually enters the brain and of where and how it might act, compounded with awareness that oxytocin has many peripheral targets at which it can act, with indirect consequences for brain activity and behaviour. The literature has been subject to intense criticism for a variety of methodological weaknesses, giving rise for example to the conclusion of some authors that, in relation to behavioral studies in man, "intranasal OT studies are generally underpowered and that there is a high probability that most of the published intranasal OT findings do not represent true effects." (Walum et al Biol Psychiatry 2016 79: 251-7). Those concerns have been amplified by concern about the integrity of an associated body of literature, measuring oxytocin in human subjects where those measurements involve an assay technique that reports values of oxytocin 100 fold higher than measures using well validated assays, and which show no correlation at all with the well-validated measures.

This study does not avoid these problems, on the contrary, it engages with them. A persistent criticism of intranasal studies has been that there has generally been no attempt to control for the peripheral consequences of intranasal oxytocin administration, although it is while known that very little of the applied large doses enter the brain, systemic concentrations are raised to unphysiologically high levels. The present study compared the effects of oxytocin given intranasally or intravenously.

The issue of peripheral effects is difficult to address comprehensively given the diversity of sites of action of oxytocin, with its known consequences for example for blood glucose and insulin concentrations, but this study does engage with one major potential target – the heart, by

combining this study with measures of heart rate variability.

Finally this study quantifies the changes in oxytocin concentration accompanying the tests by measures of plasma oxytocin using an appropriate and rigorous assay

I am aware of only one study in humans prior to the present study that has addressed this:

Quintana et al. (2019) *Neuropsychopharmacology*. 2019 Jan;44(2):306-313 compared the effects of intranasal and intravenous oxytocin in a fMRI study combined with a pupillometry study.

Importantly, that study found no significant difference between intranasal oxytocin and IV oxytocin on right amygdala activity and pupil diameter. The Quintana study used a dose of iv oxytocin designed to mimic the concentrations achieved by intranasal oxytocin; arguably that is a better design than the present study where the IV infusion achieved higher concentrations than IN administration.

The present study included a further refinement in using two different methods of intranasal administration.

The study design appears strong to me; the sample size was determined by the outcomes of a previous experiment and appears adequate in context.

I am not competent to assess the methodological rigor of the CBF and HRV analyses: the data look clear and convincing to me but this needs expert judgement. I would expect that these analyses should also be performed blind, and it would be good to have that confirmed as it is not clear to me that this was the case.

The key outcomes are that while IV and IN routes produced similar changes in CBF in some brain regions: (the amygdala and anterior cingulate cortex), the IN route alone produced changes in other brain areas. Even where IN and IV administration produced similar effect, it remains possible that these are exerted by actions of OT within the brain, given that some OT, albeit extremely small amounts, do penetrate the blood-brain barrier, but the present experiments clearly raise the possibility that these effects reflect peripheral actions. Their studies do make it unlikely that peripheral actions on the heart account for these changes, but as the authors acknowledge this is one of many potential peripheral sites of action. A possibility that also needs consideration is that IN oxytocin affects CBF directly by effects at V1 receptors on cerebral blood vessels, though such effects would also be expected to accompany IV administration.

The authors hypothesize that "the difference in the patterns of OT-induced rCBF changes achieved with each method can only be explained by differences in the deposition of OT in the olfactory and respiratory regions and the paranasal cavities which receive innervation from the olfactory and trigeminal nerves and may thus constitute important points of entry to the brain." This assumes that the IN oxytocin does not affect activity in these nerve bundles – an assumption that might be questioned given evidence that the oxytocin receptor is expressed in human taste buds (*Chem Senses*. 2014 39:359-77.) and rat olfactory epithelium (*Eur J Neurosci*. 2004 Aug;20(3):658-70.; *Neuroreport*. 2008 19:1623-6. However, these seem to me to be matters for consideration and discussion rather than additional experiments.

A particularly interesting aspect of this work is their comparison of sites of CBF change with sites of OTR and V1R expression in human brain, shown in a supplementary figure. It is not clear to me where those maps come from and how they were obtained, as the reference given seems incorrect. Very recently Quintana et al. published a map of human OTR expression in the human brain (Quintana et al. *Nat Commun*. 2019 Feb 8;10(1):668) that appears different from the map of OTR mRNA expression shown in the present paper. It would be helpful to have a much clearer understanding of how this map was generated, the potential limitations of interpretation, and the origins of any discrepancies there may be. The correspondence of regional CBF flow with sites of OTR and V1R expression is an extremely interesting issue and one that deserves expanded consideration in the present study.

Overall I found this paper clearly written, easy to follow, and in the areas that I feel competent to judge, seemed rigorous, scholarly and self critical. With the exception of the supplementary data on receptor expression, I thought that all methodological details that I expected to be present

were present. It seemed to me that the statistical approaches were appropriate for this exploratory study, transparently described and clearly documented. There are questions that could be asked about multiple comparisons, but it seems to me that these issues are quite transparent and I should leave consideration of these to the relevant experts.

Response to Reviewers' comments

Reviewer #1 (Remarks to the Author):

The attached report is of interest to the community of oxytocin researchers who need to understand the mechanisms of action of the nasal and intravenous administration. It addresses an important issue and the paper is very well written.

We thank the reviewer for acknowledging the relevance of our work and the effort we put in transmitting our message clearly.

I think the claims of this paper are overstated and it is not until one reads the end of the discussion that the authors acknowledge the limitations. This is then followed by a conclusion paragraph that returns to overstatements. They should be acknowledged much earlier and clearly in the abstract.

My main concern is the use of blood flow as an indicator of some action. It seems optimistic to infer OT action from this mechanism. Blood flow is linked to many parameters, also comprising vasopressin activation (i.e. you squeeze the vessels, you reduce the flow).

We thank the reviewer for raising this question. We think that the question the reviewer asks is: *Can measures of cerebral blood flow (CBF) be used to infer drug (in this case OT) action in the brain?*

It is important to highlight that all measures of regional CBF exploit the well-established phenomenon of neuro-vascular coupling as a means of obtaining an indirect but extremely sensitive signature of regional neuronal activity. At least two landmark publications have recently vindicated this approach:

- Spatio-temporal evolution of the Functional Magnetic Resonance Imaging Response to ultra-short stimuli. Hirano et al, Journal of Neuroscience, (2011) 1440-1447
- Resting state haemodynamics are spatio-temporally coupled to synchronised and symmetric neural activity in excitatory neurons. Ma. et al, PNAS, (2016) 113(52)E8463-E8471

Arterial spin labelling (ASL) magnetic resonance imaging is particularly well-suited for the use of regional CBF as a probe of neuronal activity because the duration of the post-labelling delay of the sequence can be accurately chosen such that the signal contribution is driven from arterioles and capillaries only. This provides a quantitative measure of regional CBF which has indeed been shown to indirectly but specifically reflect neuronal activation. Once again, numerous publications have established the sensitivity of this approach, especially when probing the effects of psychotropic compounds. For example: Selvaggi et al. (2019) and Dukart et al. (2018) showed that changes in rCBF after several different drugs not only specifically reflect the spatial distribution of the receptors for these drugs but also the drugs' affinity to their target (1, 2) .

With respect to the concern that changes in rCBF can theoretically be accounted by unspecific effects of the drug on blood vessel calibre, we would argue that this is not an issue in our study for three reasons:

- 1) As highlighted by reviewer #3, if the effects we observed after synthetic oxytocin solely reflected drug effects on vasculature (potentially mediated by the vasopressin receptor V1aR), then we should expect to see the same effects when administering OT intranasally and intravenously as both routes of administration substantially increase the concentration of the peptide in the systemic circulation to supraphysiologic levels. This is not what we found.
- 2) Such effects might have been consistent with decreases in rCBF but they are not consistent with the increases that we observe for the intranasal route. It is also worth highlighting that following intravenous administration there are no increases in rCBF whatsoever even at quite liberal statistical thresholds (not reported).
- 3) It should be noted that we report regional changes in perfusion (rCBF) after controlling for global perfusion effects. Vascular effects of the type mentioned by the reviewer would be reflected on changes in global brain perfusion, but our results did not show any significant effects of any treatment on global perfusion. In any case, specifically to account for the possibility of drug effects being mediated by vascular changes, even though these may not be significant, we conservatively

included global perfusion values as a nuisance variable in all of our analyses. We mentioned this on page 26, section “Methods – Whole-brain univariate analysis” of the original manuscript.

The idea that the drug effects we identified on rCBF do not reflect unspecific vascular effects on global perfusion is further supported by preclinical data showing that the modulation of rCBV by intranasal OT in rodents is accompanied by changes in hippocampal local field potential recordings (3). This finding further supports the neuronal origin of the changes in rCBV observed after synthetic OT administration.

In conclusion, we believe that converging evidence from multiple studies across species and modalities, the use of an intravenous comparator and the inclusion of global perfusion effects as a nuisance variable in all our analyses strongly support the conclusion that our results regarding the effects of OT on rCBF reflect specific effects on brain physiology rather than unspecific vasculature effects. We have previously mentioned this aspect in our *Introduction* – page 5 paragraph 1. However, to make this point more clearly we have now included a detailed discussion of this point in our *Discussion* – see pages 18-19. Moreover, we would like to note that no current method, in the absence of a selective OT receptor specific PET ligand or brain penetrant OT receptor antagonist for humans, would allow us to differentiate between effects on OT and V1a receptors in the brain, a point that we acknowledge and discuss in the manuscript (when interpreting the effects we observed for the nebuliser in the visual cortex, an area where the expression of OT receptor mRNA is thought to be low but the expression of V1aR seems to be higher).

Most importantly, the authors acknowledge that they do not know the exact mechanism of the reduction of blood flow, in particular if it is periphery-mediated or centrally-mediated. In lack of this, I think it is way premature to conclude that they have demonstrated that there is a nose-to-brain pathway.

The precise mechanistic pathway of the changes in rCBF cannot be determined from this study. The reviewer is correct here, but this was not the aim of the study. The aim was to demonstrate central changes in humans and investigate whether these changes can be fully recapitulated by intravenous administration of the peptide (as explained in the last two paragraphs of our *Introduction* on pages 5-6), which we have achieved. A study using selective centrally-acting oxytocin antagonist to block the effects of OT administration (compared to a non-brain penetrant peripheral antagonist) would address the reviewer’s question. This is not currently possible as a centrally-acting OT antagonist for use in humans is not currently available and questions related to selectivity have been raised for peripheral antagonists currently available for humans, such as atosiban.

Second, we quite agree that in the absence of direct evidence it is premature to conclude that we have demonstrated that there is a nose-to-brain pathway; while we do not make this claim, we are sorry if we created this impression and we are happy to revise the manuscript to ensure that the reader is not left with this impression. However, we need to emphasize that our results are strongly consistent with the existence of a nose-to-brain pathway. Otherwise, how could the specific changes that we see when we give OT intranasally, and not seen when we give OT intravenously, be explained? The support for the existence of a nose-to-brain pathway in humans derives from three converging sources of evidence: (1) the divergence of the effects produced by the intranasal and intravenous routes of administration and not by the overlap of some of the effects observed between them as alluded by the reviewer; (2) it is consistent with preclinical data (4); and (3) it is consistent with the known physiology and pharmacology of the OT system and receptor (which allow effects to differ as a function of extracellular concentration and route of entry) (5, 6).

It looks to me that the nebulizer deposits OT faster in olfactory epithelium? Is that correct? It is hard to tell from the images.

The rationale behind the use of the nebuliser in our study was the demonstrated capacity of the nebuliser to increase the deposition of molecules in the posterior and superior parts of the nasal cavity, including the olfactory epithelium, an area hypothesised to be involved in nose-to-brain transport. We highlight the advantages of the nebuliser in our methods section, where we explain how the combination of the production of small size droplet and ultrasonic vibration achieves increased deposition in the nasal cavity (7). We do not expect any meaningful differences in the *speed* with which OT is deposited to the olfactory epithelium via the two intranasal methods, we apologise if we gave this impression. We also note that our main figure 3 was too complex and may have given the wrong impression. We have replaced figure 3 with a revised version (now

included in the revised version of the manuscript as figure 3), which we believe may be easier to read and avoid this type of misunderstanding.

It is important to acknowledge a number of studies that have previously shown intravenous administration also improves social cognition (see work of Hollander)

We totally agree with the reviewer and we do mention the two studies by Hollander et al. in our *Introduction* (page 4, paragraph 1) as evidence highlighting the need to evaluate the effects of intravenous OT on brain physiology. Following the reviewer's emphasis on these important studies, we have now discussed them more explicitly in our *Discussion* (page 12, paragraph 1) as they fit and support our observations that increases in plasma OT after intravenous administration of synthetic OT modulate amygdala and anterior cingulate rCBF. Thank you for this suggestion.

It is a little unclear why the authors suggest that heart rate variability measures as the main assessment for peripheral response. There are many peripheral sites in the human body that could trigger brain release of oxytocin so the absence of an effect of OT on HRV cannot rule out other peripheral actions.

We completely agree with this point and we apologize if we gave the wrong impression that heart rate variability would account for all possible actions of synthetic OT on peripheral organs.

Heart rate information is collected routinely in our scanning protocols. Given expression of OTR on the heart muscle (8) and previously reported effects of synthetic OT on heart rate and/or heart rate variability (9, 10), studying the potential effects of increases in systemic OT on heart function as a possible mediator of changes in brain physiology is of particular interest. The absence of any such effects does not in any way imply that changes in brain physiology might not be mediated by effects of other peripheral targets. We openly acknowledge this limitation on page 18 in our *Discussion* and suggest that future studies using non-brain penetrant OT receptor antagonists will be critical to address this question.

Reviewer #2 (Remarks to the Author):

In the present work, Martins et al. explore whether different routes of oxytocin (OT) administration lead to different effects on brain activity, using arterial spin labelling as a readout. This is an important issue as a large number of studies has investigated the effects of synthetic OT on the brain, but the biological mechanisms underlying IN OT action are still unclear.

We thank the reviewer for acknowledging the relevance of our work.

However, there are two critical problems with the current manuscript, one is methodological, and the second one is interpretational.

1) The overall design and analysis are rather weak because authors choose a cluster forming threshold of $p < 0.005$ uncorrected and then analyzed the interaction time x treatment with post hoc paired t tests for each treatment condition and the interval. This means that at least 30 paired t test maps were computed (3 treatments x 10 time intervals). This high number of tests, at this permissive threshold ($p < 0.005$ uncorrected) and with 16 subjects is very likely to produce false positive results. This could explain the incoherence in the results (please see 2)).

We apologise if we gave the reviewer the impression that we report results uncorrected for multiple comparisons. From this comment, it is not clear if the reviewer takes issue with the value of the cluster forming threshold ($P < .005$) or the fact that it was "uncorrected". Therefore, our response will cover both possibilities.

Our statistical thresholds had been determined a priori and are identical to our highly cited previous work (11) and standardly applied in ASL studies measuring rCBF (12-17).

Briefly, we conducted the neuroimaging analyses using cluster level inference, using the 'Cluster Extent Criterion' (as implemented in SPM), to account for multiple comparisons, which assigned significance only to clusters that are larger than the largest cluster that can be identified by chance. This formalism is employed routinely in functional MRI and PET (18).

This substantially enhances our sensitivity to observe weaker and diffuse signals (as is particularly the case with rCBF effects) at the cost of spatial specificity. In cluster level inference, a cluster forming threshold is applied first to identify voxels with signal change of sufficient magnitude to pass at the next stage. This cluster forming threshold is typically uncorrected. There has recently been some debate regarding whether this threshold should be $P < .001$ uncorrected or higher, with all papers criticising the use of a $P < .01$ uncorrected cluster forming threshold. While there has been some controversy in the literature based on results from simulations with resting BOLD fMRI data regarding whether the cluster forming threshold should be $P < .001$ uncorrected or could be higher (19), this has been criticised in follow-up studies as possibly overly conservative and is not universally accepted (20, 21), and all of these articles focus on criteria for reporting results of BOLD fMRI investigations and certainly not in the context of ASL/rCBF studies (12-17), as it disproportionately increases Type II errors. Compared to BOLD-fMRI, ASL data present lower signal-to-noise ratio. We are not aware of any study showing that using a $p < 0.005$ cluster-forming threshold does impact on false positives discovery rate for ASL data. In fact, most concerns have been raised for highly liberal thresholds such as $p < 0.01$ (22), a threshold which we did NOT apply in our analyses. In cluster level inference correction for multiple comparisons is applied to determine the number of contiguous voxels that could not have occurred by chance, if the null hypothesis was true. This cluster size, in statistical parametric mapping, is determined using random field theory (23). All our reported clusters are significant at $p < 0.05$ using FWE-correction based on spatial cluster extent (as explained above). To this effect, we have revised a phrase in the captions of all our Tables and figures stating clearly that: "*We conducted cluster-level inference, reporting clusters significant at $p < 0.05$ FWE-corrected (cluster-forming threshold: $p < 0.005$, uncorrected)*".

In any case, given the reviewer's concerns, we did repeat our analyses using a cluster-forming threshold of $p < 0.001$ (as per stringent recommendations for BOLD-fMRI analysis). Our findings from the flexible factorial still stand for all of the effects tested. In respect to t-tests, all of our findings for the effects of nasal spray and intravenous administrations still stand. For the nebuliser, only the clusters depicting decreases in rCBF in the basal ganglia do not reach significance ($0.058 < p_{FWE} < 0.083$) in the t-tests (yet these drive and were identified in the treatment x time interaction in the flexible factorial).

Subsequently, we indeed followed up the flexible factorial model testing for treatment and time x treatment effects with a series of 24 paired sample t-tests (not 30) which we report in Tables 2, 3 and 4, providing the exact P value for every significant cluster. As reviewer #3 identifies, our study is an exploratory study – this series of paired t-tests is the backbone of our analyses as we did not have any prior expectations due to a lack of literature in this area (namely, for the potential effects of intravenous administration or administration using a nebuliser on resting rCBF changes, where our study is the first to date). Our own previous work (11) had used an entirely different and single-blind design, using two independent groups and comparing post-dosing scans to baseline scans within each group separately.

We do appreciate reviewer's #2 concern that following up on our significant treatment x interaction with 24 (3 x 8) post hoc paired-tests could raise concerns related to multiple testing. Given the exploratory nature of this study we had decided not to explicitly adjust the cluster P_{FWE} values by the number of t-tests but to allow the reader to draw their own conclusions as we presented all results and their exact P_{FWE} values in Tables 2, 3 and 4. In any case, since the reviewer thinks it is necessary to explicitly adjust the P value by the number of t-tests ($0.05/24 = 0.002$), we are happy to do so. As the reviewer can already appreciate from Tables 2, 3 and 4, most of the results of the paired sample t-tests would remain significant and the pattern of results and our interpretation would not change. For the nasal spray, only one cluster in the cerebellum/brainstem depicting decreases in rCBF 87-95 mins post-dosing would not survive. For the nebuliser, only one cluster depicting decreases in rCBF in the basal ganglia 15-23 mins post-dosing and another one depicting increases in rCBF in the cerebellum 24-32 mins post-dosing would not survive. All clusters showing decreases in rCBF when administering OT IV (compared to placebo) would survive. We have now highlighted in our tables 2, 3 and 4 all clusters that do not survive Bonferroni correction for the number of paired T-test conducted, so the reader can make his/her own conclusions. We further added a note in tables 2 – 4 that "*Clusters that do not survive*

correction for multiple testing following adjustment of the P value using Bonferroni correction for the 24 paired-T tests performed are denoted by an asterisk (*) ($P_{adjusted}=0.05/24=0.002$)”.

We would also like to note that our conclusions are further supported by our correlational analyses where we found that variations in plasma OT only predicts decreases in rCBF in the 4 clusters showing overlap between the OT-induced changes following nasal spray and intravenous administration.

There are several other minor issues with the methods and results:

- Clusters in which an interaction effect was seen (Fig3) do not match with the clusters displaying a main effect of time (Fig 2).

We are not sure we can understand this point. While the clusters in which we observed a treatment x time interaction reflect changes in rCBF after treatment which vary as a function of time-interval, the ones where we observed a main effect of time reflect clusters where rCBF changed as function of time despite the type of treatment. We have no reasons to believe these clusters should match since they are likely to reflect different phenomena: drug effects for the interaction drug x treatment; decreases in attention and/or increases in drowsiness for the main effect of time.

- Time intervals between figures, text and tables do not match (24-32min in fig3 and text, but 19-26min was used in the table?)

We apologize for this oversight and thank the reviewer for pointing this out. The time-intervals mentioned in the figure are the ones reflecting time after last drug administration offset. We have now corrected the tables accordingly, in the revised version of the manuscript.

2) The results are not coherent. Firstly, the authors do not reproduce their previous findings: in their previous work (Paloyelis et al. 2016), the authors found that "the temporal profile of IN-OT- induced rCBF changes showed a peak response 39–51 min after IN-OT, followed by a gradual diminution of effects." However, in the present article, they fail to see any effect of IN OT (nor IV and nebulizer) during this precise time period (but they do before and after, which is even more confusing). This critical issue is not mentioned in the Discussion. How could oxytocin exert effects, then it stops for 30 min and then produces other effects? Furthermore, most studies employing IN OT in humans have precisely used this 45-75 min post administration interval to test their subjects and to find modifications of brain activity, which contradicts the absence of effects at this time interval in the present study.

We thank the reviewer for raising these important points that we had not discussed in sufficient clarity in the manuscript.

First, we would like to indicate that our own previous work (11) that the reviewer refers to had used an entirely different and single-blind design, using two independent groups and comparing post-dosing scans to baseline scans within each group separately. The analyses that the reviewer refers to used pattern recognition to estimate the temporal dynamics of the effects of intranasal OT. Pattern recognition has the advantage of improving sensitivity to small effects and is driven by the overall pattern of changes – where each voxel contributes to the classification result (according to the voxel’s weight). This technique cannot and should not be used to make inferences regarding the location of a drug’s effects. In that paper (11), we also conducted univariate analyses, by comparing an average of all post-dosing scans to the baseline scans. These analyses did show rCBF changes (compared to baseline) that overlap with the areas we identify here, e.g. frontal gyrus, insula, cingulate cortex, cerebellum, hippocampus, amygdala.

Our current study built on that earlier proof of concept work following up with a design that is considered a gold-standard in pharmacological studies in humans: a randomised, double-blind, cross over design, which increases power by minimizing inter-individual differences between active and placebo conditions. Importantly, our design and reported analyses match the questions we sought to address in this study: can the effects on rCBF observed after intranasal OT be explained by increases of OT in systemic circulation? Our purpose has not been to ascertain the temporal dynamics of the effects, hence we do not make such claims. It

possible that the signal-to-noise of ASL data in some of the intermediate time-points was lower than could be detected at our predefined significance threshold in individual t-tests.

We should also note that it is plausible that the temporal dynamic of the effects are locally different, that is, that they vary according to OT receptor distribution and density and as a function of extracellular concentration (for example, this might determine whether the effects reflect direct receptor targeting or engagement of other neurochemical systems such as V1aR).

In our opinion, given what we know and speculate about the physiology and pharmacology of the OT system in the brain (24, 25), we cannot assume that synthetic OT modulates rCBF in all areas with the same temporal dynamics. In fact, ASL studies in rodents (3) looking at the effects of intranasal OT on CBV responses to synthetic OT are compatible with the existence of two-waves of functional responses reflecting the co-existence of acute/transient with sustained effects. We have now accommodated these points in the revised version of the manuscript (*Discussion* – page 19), highlighting that inferences on the temporal dynamics of the effects we report should not be made.

Secondly, it is hardly conceivable that the nebulizer produces effects completely different than the spray or intravenous administration. The nebulizer optimizes the amount of OT delivered to the posterior regions of the nasal cavity, but it does not permit OT to access new regions. Thus, it should lead to stronger effects than IN OT and, at the very least, we should see some overlap between the nebulizer and the spray effects. This incoherence is present throughout the discussion: authors first state that some of the IN OT effects are mediated by the increase of plasma OT, while some other effects (increase in rCBF) are mediated by a direct nose to brain pathway. This indicates, again, that IN OT effects should be in accordance with the ones obtain after nebulizer administration. The authors' explanation that OTR binds to different G protein is extremely far stretched, as the distribution of OTR is still very unclear in the human brain, so inferring regional differences in G protein binding is purely theoretical, and does not explain a complete mismatch between regions modulated by the spray and the nebulizer.

The reviewer raises valid and insightful points here.

As we state at the end of our introduction, our initial hypothesis was indeed that the nebuliser would result in a more robust pattern of changes that would be comparable to the spray but of higher magnitude and eventually include areas that could not be targeted neither by the spray nor by the intravenous administrations. However, we should acknowledge that this prediction would mostly be valid if the pharmacodynamics of rCBF response to OT followed a linear model – which does not seem to be the case according to some studies which argue that in fact the response in terms of brain function may actually follow an inverted-U shape function (24). While the spray and the nebuliser do not differ in the amounts of OT absorbed into the peripheral circulation, the changes we observed in terms of rCBF seem to be different, an observation we speculate may be related to differences in the amounts of OT absorbed centrally between these 2 intranasal methods (based on the established higher deposition in putative points of entry with the nebuliser). Assuming that the pharmacodynamics model follows, at least, for some areas (such as the amygdala) an inverted-U shape function, higher central bioavailability may in fact result in effects of lower magnitude/absence of effects, rather than effects of higher magnitude. Differences in the levels of coupling of the OTR with stimulatory or inhibitory G-proteins as a function of central bioavailability of OT (25) may parsimoniously account for this inverted-U shape function – however testing this hypothesis is virtually impossible in humans. We make this point now clearer in our *Discussion* – paragraph 1, page 15).

In our discussion, we acknowledge this hypothesis is speculative (*Discussion* – end of paragraph 1, page 15) and engage with this aspect by calling for further dose-response studies using the nebuliser. While these differences are puzzling, we do not believe this aspect undermines the main conclusions of our study.

Reviewer #3 (Remarks to the Author):

We thank the reviewer for the positive appreciation of our study and its design, which they find strong, their view that our statistical approach has been appropriate for this exploratory study, transparently described and clearly documented, and for finding our manuscript rigorous, scholarly and self-critical.

This paper “Do direct nose-to-brain pathways underlie intranasal oxytocin-induced changes in regional cerebral blood flow in humans?” bravely enters a veritable minefield. In recent years there has been a massive proliferation of interest in the brain oxytocin system and its role in not only social behavior but also appetite regulation and pain processing. Much of that interest has been engaged by a now vast number of studies in humans using intranasal application of oxytocin. However despite the volume of published work there remains at best a minimal understanding of how much oxytocin actually enters the brain and of where and how it might act, compounded with awareness that oxytocin has many peripheral targets at which it can act, with indirect consequences for brain activity and behaviour. The literature has been subject to intense criticism for a variety of methodological weaknesses, giving rise for example to the conclusion of some authors that, in relation to behavioral studies in man, “intranasal OT studies are generally underpowered and that there is a high probability that most of the published intranasal OT findings do not represent true effects.” (Walum et al Biol Psychiatry 2016 79: 251-7). Those concerns have been amplified by concern about the integrity of an associated body of literature, measuring oxytocin in human subjects where those measurements involve an assay technique that reports values of oxytocin 100 fold higher than measures using well validated assays, and which show no correlation at all with the well-validated measures. This study does not avoid these problems, on the contrary, it engages with them. A persistent criticism of intranasal studies has been that there has generally been no attempt to control for the peripheral consequences of intranasal oxytocin administration, although it is while known that very little of the applied large doses enter the brain, systemic concentrations are raised to unphysiologically high levels. The present study compared the effects of oxytocin given intranasally or intravenously. The issue of peripheral effects is difficult to address comprehensively given the diversity of sites of action of oxytocin, with its known consequences for example for blood glucose and insulin concentrations, but this study does engage with one major potential target – the heart, by combining this study with measures of heart rate variability. Finally this study quantifies the changes in oxytocin concentration accompanying the tests by measures of plasma oxytocin using an appropriate and rigorous assay.

I am aware of only one study in humans prior to the present study that has addressed this: Quintana et al. (2019) Neuropsychopharmacology. 2019 Jan;44(2):306-313 compared the effects of intranasal and intravenous oxytocin in a fMRI study combined with a pupillometry study. Importantly, that study found no significant difference between intranasal oxytocin and IV oxytocin on right amygdala activity and pupil diameter. The Quintana study used a dose of iv oxytocin designed to mimic the concentrations achieved by intranasal oxytocin; arguably that is a better design than the present study where the IV infusion achieved higher concentrations than IN administration. The present study included a further refinement in using two different methods of intranasal administration. The study design appears strong to me; the sample size was determined by the outcomes of a previous experiment and appears adequate in context.

We thank the reviewer for these thoughts. Just to note that it was by design that the IV comparator achieves consistently higher concentrations than the nasal administration methods throughout the observation interval. Our purpose was to conduct a proof of concept study. Since with even such a large IV OT dosage, and no matter how much we lower the statistical threshold, we do not see any OT-induced increases in rCBF when administered intravenously, compared to placebo, this suggests that the increases in rCBF that we observe after administering OT intranasally cannot be attributed to changes in plasma OT levels at all, which strengthens support for a nose-to-brain pathway.

We would like to note that in the study by Quintana et al. (the amygdala response was reported in Psychoneuroendocrinology, Volume 69, July 2016, Pages 180-188), which is a task-based study looking at amygdala activity 40-60 min post-dosing, the reported data do not fully support the authors' conclusion that “dampening of the amygdala activity in response to emotional stimuli occurs via direct intranasal delivery pathways”, as their data show dampening of amygdala activity both when OT is administered intranasally with the Breath Powered device and intravenously (compared to placebo, for happy and ambiguous faces, Fig. 4) – which is exactly what our study unambiguously demonstrates at rest and at an earlier interval.

Finally, we would like to highlight that our study is novel in multiple ways: it is the first study to investigate the pharmacodynamics effects of synthetic oxytocin on resting brain physiology over an extended period of time when administered with any of the three methods of administration we used in a double-blind placebo-controlled crossover design. It is also the first in man study regarding the effects of OT administered intravenously or with a nebuliser on resting rCBF. To make sure the contribution we make to the field is entirely clear to the reader, we have now stated explicitly the novelty of our study in our *Methods* (paragraph 2, page 28) and *Discussion* (paragraph 1, page 11).

I am not competent to assess the methodological rigor of the CBF and HRV analyses: the data look clear and convincing to me but this needs expert judgement. I would expect that these analyses should also be performed blind, and it would be good to have that confirmed as it is not clear to me that this was the case.

We thank the reviewer for his point and the opportunity to clarify that our analyses were not performed blind – which we will add in the revised version of the manuscript. Since we used a priori and commonly accepted statistical threshold as previously employed in our work (26) and numerous other papers measuring pharmacological effects or rCBF (12-17) and we report all observed results at this threshold we believe that there is little if any room for bias. We have now stated this aspect explicitly at the end of our *Methods* section on page 31.

The key outcomes are that while IV and IN routes produced similar changes in CBF in some brain regions: (the amygdala and anterior cingulate cortex), the IN route alone produced changes in other brain areas. Even where IN and IV administration produced similar effect, it remains possible that these are exerted by actions of OT within the brain, given that some OT, albeit extremely small amounts, do penetrate the blood-brain barrier, but the present experiments clearly raise the possibility that these effects reflect peripheral actions. Their studies do make it unlikely that peripheral actions on the heart account for these changes, but as the authors acknowledge this is one of many potential peripheral sites of action. A possibility that also needs consideration is that IN oxytocin affects CBF directly by effects at V1 receptors on cerebral blood vessels, though such effects would also be expected to accompany IV administration.

The authors hypothesize that “the difference in the patterns of OT-induced rCBF changes achieved with each method can only be explained by differences in the deposition of OT in the olfactory and respiratory regions and the paranasal cavities which receive innervation from the olfactory and trigeminal nerves and may thus constitute important points of entry to the brain.” This assumes that the IN oxytocin does not affect activity in these nerve bundles – an assumption that might be questioned given evidence that the oxytocin receptor is expressed in human taste buds (Chem Senses. 2014 39:359-77.) and rat olfactory epithelium (Eur J Neurosci. 2004 Aug;20(3):658-70.; Neuroreport. 2008 19:1623-6. However, these seem to me to be matters for consideration and discussion rather than additional experiments.

We thank the reviewer for bringing these studies to our attention. They indeed provide additional mechanisms through which intranasally administered OT might be affecting brain function. It is our understanding that if direct modulation of activity in these nerve bundles would account for all the changes we report for our intranasal administration methods, then one would expect that the changes observed in rCBF after intranasal administration of synthetic OT would be mostly restricted to the olfactory/trigeminal pathways and respective connected areas, which is not entirely supported by our data (for example, we have little evidence to believe that modulation of activity in olfactory or trigeminal nerves would result in changes in rCBF in the frontal gyrus). We would like to highlight that in fact most of the changes we report map to areas where expression of receptors for oxytocin (either the OTR or the AVPR1A) seem to be present, as per our current understanding of the distribution of these receptors in the human brain. It is an empirical question to what extent these mechanisms might account for at least some of the observed changes we report for the nasal spray and nebuliser in terms of central effects. At the moment, we only know that the nebuliser achieves higher deposition in the olfactory region than the nasal spray (7) – as well as that the nebuliser and nasal spray result in similar increases of OT concentration in saliva (results not presented here). We have now included this aspect in our *Discussion* (page 16, paragraph 1).

A particularly interesting aspect of this work is their comparison of sites of CBF change with sites of OTR and V1R expression in human brain, shown in a supplementary figure. It is not clear to me where those maps

come from and how they were obtained, as the reference given seems incorrect. Very recently Quintana et al. published a map of human OTR expression in the human brain (Quintana et al. Nat Commun. 2019 Feb 8;10(1):668) that appears different from the map of OTR mRNA expression shown in the present paper. It would be helpful to have a much clearer understanding of how this map was generated, the potential limitations of interpretation, and the origins of any discrepancies there may be. The correspondence of regional CBF flow with sites of OTR and V1R expression is an extremely interesting issue and one that deserves expanded consideration in the present study.

We apologize if our description of how these maps were generated runs short in details. The reference we cite is indeed inaccurate and should correspond to the methodological paper describing how such maps can be generated for almost all genes (including the OT and vasopressin receptor genes) included in the Allen Brain Atlas repository (27). These maps are freely accessible in an open repository, which we used to retrieve them. Briefly, while both Quintana et al. 2019 and Gryglewski et al. 2018 use the same data, the latter applies a variogram model in Gaussian process regression allowing to infer gene expression with high resolution and across the whole brain (results validated against PET mapping), while Quintana et al. provides a simple representation of a sparse number of areas sampled in the left hemisphere (this approach is inherently biased by the non-uniform distribution of the available samples and does not take advantage of the samples collected in the right hemisphere – which although in small number can be informative). In addition, Quintana et al. 2019 does not provide maps for the distribution of the AVPR1A gene, which we think can also provide interesting insights to interpret our findings. We are happy that the reviewer agrees that these maps provide insights that can help us interpret our findings. We have now added a brief explanation of the methods underlying the generation of these maps (*Supplementary – Fig.S4*) and referenced to the paper by Quintana et al. (2019) in the revised version of the manuscript (Ref. 71).

Overall I found this paper clearly written, easy to follow, and in the areas that I feel competent to judge, seemed rigorous, scholarly and self critical. With the exception of the supplementary data on receptor expression, I thought that all methodological details that I expected to be present were present. It seemed to me that the statistical approaches were appropriate for this exploratory study, transparently described and clearly documented. There are questions that could be asked about multiple comparisons, but it seems to me that these issues are quite transparent and I should leave consideration of these to the relevant experts.

(signed; Gareth Leng)

Please see response to reviewer #2 with respect to the multiple comparisons concern.

References

1. Selvaggi P, Hawkins PCT, Dipasquale O, Rizzo G, Bertolino A, Dukart J, et al. Increased cerebral blood flow after single dose of antipsychotics in healthy volunteers depends on dopamine D2 receptor density profiles. *Neuroimage*. 2019;188:774-84.
2. Dukart J, Holiga S, Chatham C, Hawkins P, Forsyth A, McMillan R, et al. Cerebral blood flow predicts differential neurotransmitter activity. *Sci Rep*. 2018;8(1):4074.
3. Galbusera A, De Felice A, Girardi S, Bassetto G, Maschietto M, Nishimori K, et al. Intranasal Oxytocin and Vasopressin Modulate Divergent Brainwide Functional Substrates. *Neuropsychopharmacology*. 2017;42(7):1420-34.
4. Beard R, Singh N, Grundschober C, Gee AD, Tate EW. High-yielding (18)F radiosynthesis of a novel oxytocin receptor tracer, a probe for nose-to-brain oxytocin uptake in vivo. *Chem Commun (Camb)*. 2018;54(58):8120-3.
5. Grinevich V, Knobloch-Bollmann HS, Eliava M, Busnelli M, Chini B. Assembling the Puzzle: Pathways of Oxytocin Signaling in the Brain. *Biol Psychiatry*. 2016;79(3):155-64.
6. Busnelli M, Sauliere A, Manning M, Bouvier M, Gales C, Chini B. Functional selective oxytocin-derived agonists discriminate between individual G protein family subtypes. *J Biol Chem*. 2012;287(6):3617-29.
7. Xi J, Yuan JE, Zhang Y, Nevorski D, Wang Z, Zhou Y. Visualization and Quantification of Nasal and Olfactory Deposition in a Sectional Adult Nasal Airway Cast. *Pharm Res*. 2016;33(6):1527-41.

8. Shojo H, Kaneko Y. Characterization and expression of oxytocin and the oxytocin receptor. *Mol Genet Metab.* 2000;71(4):552-8.
9. Kemp AH, Quintana DS, Kuhnert RL, Griffiths K, Hickie IB, Guastella AJ. Oxytocin increases heart rate variability in humans at rest: implications for social approach-related motivation and capacity for social engagement. *PLoS One.* 2012;7(8):e44014.
10. Tracy LM, Gibson SJ, Labuschagne I, Georgiou-Karistianis N, Giummarra MJ. Intranasal oxytocin reduces heart rate variability during a mental arithmetic task: A randomised, double-blind, placebo-controlled cross-over study. *Prog Neuropsychopharmacol Biol Psychiatry.* 2018;81:408-15.
11. Paloyelis Y, Doyle OM, Zelaya FO, Maltezos S, Williams SC, Fotopoulou A, et al. A Spatiotemporal Profile of In Vivo Cerebral Blood Flow Changes Following Intranasal Oxytocin in Humans. *Biol Psychiatry.* 2016;79(8):693-705.
12. Mutsaerts H, Mirza SS, Petr J, Thomas DL, Cash DM, Bocchetta M, et al. Cerebral perfusion changes in presymptomatic genetic frontotemporal dementia: a GENFI study. *Brain.* 2019;142(4):1108-20.
13. Takeuchi H, Taki Y, Hashizume H, Sassa Y, Nagase T, Nouchi R, et al. Cerebral blood flow during rest associates with general intelligence and creativity. *PLoS One.* 2011;6(9):e25532.
14. Joe AY, Tielmann T, Bucerius J, Reinhardt MJ, Palmedo H, Maier W, et al. Response-dependent differences in regional cerebral blood flow changes with citalopram in treatment of major depression. *J Nucl Med.* 2006;47(8):1319-25.
15. Thomas BP, Yezhuvath US, Tseng BY, Liu P, Levine BD, Zhang R, et al. Life-long aerobic exercise preserved baseline cerebral blood flow but reduced vascular reactivity to CO₂. *J Magn Reson Imaging.* 2013;38(5):1177-83.
16. Loggia ML, Kim J, Gollub RL, Vangel MG, Kirsch I, Kong J, et al. Default mode network connectivity encodes clinical pain: an arterial spin labeling study. *Pain.* 2013;154(1):24-33.
17. Nwokolo M, Amiel SA, O'Daly O, Byrne ML, Wilson BM, Pernet A, et al. Hypoglycemic thalamic activation in type 1 diabetes is associated with preserved symptoms despite reduced epinephrine. *J Cereb Blood Flow Metab.* 2019;271678X19842680.
18. Friston KJ, Worsley KJ, Frackowiak RS, Mazziotta JC, Evans AC. Assessing the significance of focal activations using their spatial extent. *Hum Brain Mapp.* 1994;1(3):210-20.
19. Eklund A, Nichols TE, Knutsson H. Cluster failure: Why fMRI inferences for spatial extent have inflated false-positive rates. *Proc Natl Acad Sci U S A.* 2016;113(28):7900-5.
20. Lieberman MD, Cunningham WA. Type I and Type II error concerns in fMRI research: re-balancing the scale. *Soc Cogn Affect Neurosci.* 2009;4(4):423-8.
21. Slotnick SD. Resting-state fMRI data reflects default network activity rather than null data: A defense of commonly employed methods to correct for multiple comparisons. *Cogn Neurosci.* 2017;8(3):141-3.
22. Eklund A, Knutsson H, Nichols TE. Cluster failure revisited: Impact of first level design and physiological noise on cluster false positive rates. *Hum Brain Mapp.* 2019;40(7):2017-32.
23. Flandin G, Friston KJ. Analysis of family-wise error rates in statistical parametric mapping using random field theory. *Hum Brain Mapp.* 2019;40(7):2052-4.
24. Spengler FB, Schultz J, Scheele D, Essel M, Maier W, Heinrichs M, et al. Kinetics and Dose Dependency of Intranasal Oxytocin Effects on Amygdala Reactivity. *Biological Psychiatry.* 2017;82(12):885-94.
25. Busnelli M, Chini B. Molecular Basis of Oxytocin Receptor Signalling in the Brain: What We Know and What We Need to Know. *Curr Top Behav Neurosci.* 2018;35:3-29.
26. Paloyelis Y, Doyle OM, Zelaya FO, Maltezos S, Williams SC, Fotopoulou A, et al. A Spatiotemporal Profile of In Vivo Cerebral Blood Flow Changes Following Intranasal Oxytocin in Humans. *Biological Psychiatry.* 2016;79(8):693-705.
27. Gryglewski G, Seiger R, James GM, Godbersen GM, Komorowski A, Unterholzner J, et al. Spatial analysis and high resolution mapping of the human whole-brain transcriptome for integrative analysis in neuroimaging (vol 176, pg 259, 2018). *Neuroimage.* 2019;188:845-.

Reviewers' Comments:

Reviewer #1:

Remarks to the Author:

The authors provide a case that OT drug administration induces rCBF changes in the brain. Just as one might observe with other imaging studies, these changes are of interest to the international readership. The pathway and what these changes actually mean is the debate that the authors and I disagree with.

The specific question raised in the title and throughout the paper is whether this paper informs on the pathway of absorption from nasal or intravenous delivery. The authors rely on correlations of effect to infer how that effect occurs (causation). The data cannot support causation.

I do not wish to offend the authors, but in my opinion there is actually no data in this manuscript that tests the mechanism or nose-to-brain pathway. While in the response letter, the authors say they do not make the claim, the claim is inferred in the title and repeated throughout the manuscript and is the reason why this paper may be viewed as particularly significant.

The authors make the following statement, as an example:

'The use of an intravenous comparator can illuminate whether intranasal OT induced changes on brain perfusion in humans are associated with nose-to-brain pathways or result from concomitant increases in systemic OT circulation.'

The statement implies that we understand the mechanisms of absorption from either intravenous or intranasal mechanisms and therefore the two should be compared to compare the pathways, when the reality is we actually do not understand the absorption pathways of either.

I don't have further comment beyond this as my belief is the real significance of this data is in showing a novel neural effect following OT administration but this novel neural effect does not inform on how the effect was caused.

Reviewer #2:

Remarks to the Author:

I appreciate authors' work on the revision of the statistical analysis (e.g., correcting T test for multiple comparisons).

However, there are few remaining concerns:

1) The authors stated that their technique should not be used to infer temporal aspects of the response, yet all their results are based on temporal segmentation of the data (there are no effects when time factor is not in the analysis, meaning no main effect of treatment). Most of previous studies in animals and human demonstrated that IN OT effects start after 20-30 min, last for about an hour, peaking at 45 min, and then decrease back to baseline at 90 min. Can the authors comment on their results on the appearance of effects between 15 to 40 min, followed by no effects for 30 min and the appearance of a novel effect at 75 min?

2) The mismatch between spray and nebulizer. The authors proposed non-linear effects of OT (inverted U shape theory), but finding no common effects is surprising. The administration of the huge amount of OT may saturate potential nose-to-brain pathway (if existing). Can the authors comment on this issue?

3) The author may cite and probably compare their results with recently published results from Scott Young lab: Smith et al. "Oxytocin delivered nasally or intraperitoneally reaches the brain and

plasma of normal and oxytocin knockout mice". Pharmacology research, 2019.

Reviewer #3:

Remarks to the Author:

The authors have responded carefully and appropriately to all of the issues that I raised, and as far as I can see to the other points also.

There are certainly statements in this paper that I would disagree with and I remain skeptical about some of the interpretation. However, this study is by the standards of the field, exceptionally rigorous, carefully controlled and the interpretation balanced and thoughtful.

Including intravenous oxytocin infusions as a control is an important and practical step to include in future behavioral and brain imaging studies using oxytocin: intravenous delivery will be more reliable, controllable and reproducible than intranasal delivery, and clearly there is at least very considerable overlap in its consequences for brain activity. However it should be much easier using IV delivery to establish exactly where the primary sites of action are _ including the key question of whether these are peripheral or central.

Reviewer #4:

Remarks to the Author:

The article by Martins et al "Do direct nose-to-brain pathways underlie intranasal oxytocin-induced changes in regional cerebral blood flow in humans?" tackles a controversial issue and has triggered some controversial reviews. In reading the article and the accompanying comments, I will only add in comments as they relate to my areas of competency: perfusion imaging and functional MRI analysis. I remain agnostic to the argument for separate pathways between the nose and the brain, circumventing the BBB. I hope my comments will be useful to the authors for a re-analysis and a stronger resubmission of this article, which tackles an interesting and innovative question.

Minor comments:

P5: "...As a result of neuro-vascular coupling, changes in rCBF are likely to reflect changes in neuronal activity rather than vascular effects..." by definition, rCBF methods would have to capture vascular effects as well as neural activity. Remove the clause "rather than vascular effects"

Figure 4: the legend needs another caption/line denoting the color used for the overlapping of regions.

Page 15: The sentences "... Our initial hypothesis was that the nebuliser would result in a more robust pattern of changes that would be comparable to the spray but of higher magnitude and eventually include areas that could not be targeted neither by the spray nor by the intravenous administrations. However, we should acknowledge that this prediction would have mostly been valid if the pharmacodynamics of the rCBF response to OT followed a linear model – which does not seem to be the case at least for some brain areas..." seem like run-on sentences and are very difficult to follow. Please edit accordingly.

Major comments:

The strategy of analysing the data a large number of t-tests seems like a poor use of the data and invites problems with multiple comparisons, as discussed. Accordingly, this approach yields a temporal rCBF evolution that doesn't make a lot of sense, as pointed out by another of the reviewers.

Given that there is no a priori model for the temporal evolution of the rCBF, there are better

options. They could easily use PCA or ICA to search for patterns in the time course.

Additionally, I think it is crucial to extract the rCBF time courses (8 time points) from preconstructed ROIs and try to generate a kinetic model based on those. The plasma concentration of OT should be part of the model. I note that the 8 minute ASL acquisitions had many averages. For the purpose of time series analysis and kinetic modelling, it is more efficient to NOT average those images, but rather use a larger number of noisier time points. The authors have 5 pairs of images collected 8 times. Using a surround subtraction scheme, they could be 38 time points instead.

What is the correlation between plasma OT and rCBF? Those correlation maps would be very helpful in settling the debate.

Page 16: The authors argue "... If direct modulation of activity in these nerve bundles would account for all the changes we report for our intranasal administration methods, then one would expect that the changes observed in rCBF after intranasal administration of synthetic OT would be mostly restricted to the olfactory/trigeminal pathways and respective connected areas, which is not entirely supported by our data (for example, we have little evidence to believe that modulation of activity in olfactory or trigeminal nerves would result in changes in rCBF in the frontal gyrus)... ". It is just as likely that the authors have detected an unexpected connectivity pathway as that they have detected a nose-brain transport system.

The authors are careful to remove global CBF effects. The global CBF time course may contain valuable information about the effect of OT and accounts for the differences between administration routes and would deserve its own analysis as well as serve as a regressor of no interest..

Reviewers' comments:

Reviewer #1 (Remarks to the Author):

The authors provide a case that OT drug administration induces rCBF changes in the brain. Just as one might observe with other imaging studies, these changes are of interest to the international readership. The pathway and what these changes actually mean is the debate that the authors and I disagree with.

The specific question raised in the title and throughout the paper is whether this paper informs on the pathway of absorption from nasal or intravenous delivery. The authors rely on correlations of effect to infer how that effect occurs (causation). The data cannot support causation.

I do not wish to offend the authors, but in my opinion there is actually no data in this manuscript that tests the mechanism or nose-to-brain pathway. While in the response letter, the authors say they do not make the claim, the claim is inferred in the title and repeated throughout the manuscript and is the reason why this paper may be viewed as particularly significant.

The authors make the following statement, as an example:

'The use of an intravenous comparator can illuminate whether intranasal OT induced changes on brain perfusion in humans are associated with nose-to-brain pathways or result from concomitant increases in systemic OT circulation.'

The statement implies that we understand the mechanisms of absorption from either intravenous or intranasal mechanisms and therefore the two should be compared to compare the pathways, when the reality is we actually do not understand the absorption pathways of either.

I don't have further comment beyond this as my belief is the real significance of this data is in showing a novel neural effect following OT administration but this novel neural effect does not inform on how the effect was caused.

We thank the reviewer for his/her comments and we note their difference of opinion without taking any offence. Science is a domain where differences of opinion should be encouraged and can be constructive. Neuroimaging is indeed a correlational method and cannot inform on the exact mechanisms of absorption. We apologise if our wording inadvertently may have given the impression that this is what we claim, and we thank the reviewer for highlighting this point.

We would like to note though that our *study design* is not correlational. We believe that our study design allows us to address our aims and draw two key conclusions: (1) that increases in plasmatic oxytocin do induce certain changes in brain perfusion without the involvement of nose-to-brain pathways (as attested by the observation of oxytocin-induced changes in perfusion when oxytocin was administered intravenously); (2) increases in plasmatic oxytocin cannot account for all changes in brain perfusion observed following the intranasal administration in oxytocin, which is consistent with the existence of nose-to-brain pathways (but does not prove their existence, or illuminate the exact mechanism – these were not aims that could have been or were intended to be addressed with the current study design).

Noting the reviewer's concerns, we have revised or toned down where possible statements to avoid giving the wrong impression. For instance, we revised the statement the reviewer mentioned above to read:

"The use of an intravenous comparator can illuminate whether intranasal OT-induced changes on brain perfusion in humans result from concomitant increases in systemic OT circulation ~~or could have a contribution from nose-to-brain pathways.~~" (page 5, paragraph 3)

And:

"While our findings are consistent with the idea that nose-to-brain pathways ~~could contribute to some of the changes in rCBF induced by intranasal OT, our study cannot provide evidence regarding the precise mechanisms underlying these effects.~~" (page 17)

We also implemented further revisions of the title of the manuscript, abstract and on pages 5, 17 and 21 (see highlighted text).

Reviewer #2 (Remarks to the Author):

I appreciate authors' work on the revision of the statistical analysis (e.g., correcting T test for multiple comparisons).

However, there are few remaining concerns:

1) The authors stated that their technique should not be used to infer temporal aspects of the response, yet all their results are based on temporal segmentation of the data (there are no effects when time factor is not in the analysis, meaning no main effect of treatment). Most of previous studies in animals and human demonstrated that IN OT effects start after 20-30 min, last for about an hour, peaking at 45 min, and then decrease back to baseline at 90 min. Can the authors comment on their results on the appearance of effects between 15 to 40 min, followed by no effects for 30 min and the appearance of a novel effect at 75 min?

2) The mismatch between spray and nebulizer. The authors proposed non-linear effects of OT (inverted U shape theory), but finding no common effects is surprising.

We thank the reviewer for highlighting these concerns. We will respond to both concerns above together, as they can be both addressed by a further analysis that we now present (which was also undertaken in response to a request by reviewer 4).

Indeed, we agree with the reviewer that the temporal *segmentation* of oxytocin-induced changes in rCBF (implemented by examining each time interval separately) has been at the core of our analyses. We believe there were good reasons to justify our approach. Given the long temporal interval that we sampled post-dosing, and the previous evidence suggesting the temporal specificity of oxytocin effects (some of which the reviewer discusses), we thought that a main effect of treatment would be unlikely. We did though predict and observe a treatment x method interaction – suggesting the temporal specificity of

oxytocin effects. To map this temporal specificity in time and in space we examined each 8 min time interval separately so as to ascertain the pharmacodynamics of oxytocin with relative temporal and spatial precision, as we expected the effects to be time and region specific (unpublished univariate analyses, for each separate time interval, on our 2016 paper in *Biological Psychiatry*(1) suggested the temporal dynamics of oxytocin-induced rCBF changes varied by region and time).

However, it entailed certain disadvantages as well. ASL is characterised by low signal to noise ratio (the narrower the sampled interval, the less the number of control-label pairs, the lower the signal to noise ratio). Hence, even though our study was generally adequately powered, by focusing on each time interval separately could result in two undesired consequences. First, threshold effects: we may not see rCBF changes that fall short of our statistical threshold, not because there is no oxytocin-induced rCBF change at the specific time interval (relative to the time intervals before and after), but because the noise may happen to be higher at that interval. The second unwanted consequence follows from this and was also highlighted by the reviewer: inadvertently, the reader might draw inferences about the *temporal dynamics* of oxytocin-induced changes in rCBF, which would not necessarily be justified given our analyses and the points above.

Thanks to this reviewer's and reviewer's 4 advice, to address these issues in the context of our existing analyses, we have now conducted a more detailed examination of the temporal dynamics of the effects we observed for the three different methods of administration we used herein, focusing on specific key anatomical regions. We believe this makes better justice to our data – always within the constraints of our existing analyses (that is, always being mindful that our analyses are not circular).

Specifically, we added the following analysis (as described on pages 31/32 in the Materials and Methods section):

Regional temporal dynamics of pharmacological effects

We explored the time-course of rCBF changes by estimating the mean rCBF values using the `fslmeans` command in anatomical regions of interest (ROIs) corresponding to clusters showing significant OT-induced rCBF changes (compared to placebo) for each method of administration in the whole brain univariate analyses. Specifically, we defined anatomical ROIs for the left amygdala, anterior cingulate gyrus, left caudate, left putamen, left globus pallidum, right visual cortex, left posterior insula and left superior frontal gyrus using the Harvard-Oxford cortical and subcortical atlases distributed with FSL. To maximize the amount of data points used to fit the time-course functions (from 8 to 40 time points), at a cost of including noisier estimates, we estimated mean ROI rCBF values of each of the 5 rCBF maps corresponding to the 5 C-L pairs that were averaged to obtain the rCBF map for each of the 8 time-intervals. We estimated the rCBF time-course for each ROI implementing a non-parametric regression method that fits multiple regressions in local neighbourhood (Locally estimated scatterplot smoothing - LOESS) using the `ggplot2` package from R (109). This approach allows us to capture possible non-linear relationships between rCBF and time, while reducing the noise and without making any assumptions about the relationship between these two variables. To control for the potential confounding effects of the small changes in global CBF across time or between methods of administration, we applied the LOESS method to adjust ROI mean rCBF estimates for global CBF, which corresponds to estimating raw ROI mean rCBF minus global CBF values. Since we performed these informative analyses a posteriori following the advice of one of our reviewers, we did not run any statistical analysis on these extracted data, to avoid double dipping. Instead, we provide these time-courses to illustrate the temporal

dynamics of OT-induced rCBF changes, by method of administration, for each key brain region showing significant OT-induced changes in rCBF in the univariate whole brain analyses.

We provide a new figure (Figure 6) and comment on these analyses in Results (page 11):

Regional temporal dynamics of pharmacological effects

We present in Fig. 6 graphs depicting changes over time in regional perfusion in anatomical regions of interest (ROIs) corresponding to clusters showing significant OT-induced rCBF changes (compared to placebo) for each method of administration in the whole brain univariate analyses. Overall, these graphs show that the temporal dynamics of the effects of synthetic oxytocin administration on brain perfusion is complex and shows regional specificities. Depending on the region of the brain, the effects can start or be maximal at different times post-dosing and extend over different periods of time. For instance, Figure 6 shows that the OT-induced decrease in left amygdala perfusion is largely an early effect, maximal in the first few minutes (< 25min) after OT administration and independent of method of administration. In contrast, the OT-induced rCBF increase observed in the posterior insula after administration by standard nasal spray tends to be more sustained in time and maximal at later time-intervals.

and further comment in Discussion (see highlighted parts on pages 16, 18 and 21).

We believe that the new insights that this analysis brings to the manuscript helps clarify/address the concerns raised by the reviewer and make better justice to our data. For example, they show that the absence of effects in intermediate time intervals when such effects are observed before and after, is likely to be due to statistical thresholding (e.g. resulting from greater variance at a specific interval, rather than a true absence of effect). Indeed, the plotted rCBF values by time shown in figure 6 suggest that oxytocin-induced changes in the cluster including the caudate nucleus, when oxytocin is administered with the nebuliser, are consistently lower than other administration methods across time intervals (but fluctuations in rCBF values over time in the placebo condition may result in this effect appearing as significant, at a given threshold, only at specific time intervals). In contrast, figure 6 also illustrates that oxytocin-induced changes in other brain regions are specific to specific time intervals. For example, the decrease in rCBF in the left amygdala is maximal <25 min post-dosing, *across methods* (and may even peak before 15 min post-dosing, which falls outside the temporal interval we sampled rCBF in our studies). In the right visual cortex, the increase in rCBF is maximal between 25-50 min.

In sum, we believe our approach served our aims well. It provided specific information about the temporal and spatial specificity of oxytocin-induced changes in rCBF as a function of method of administration, which previously lacked in the literature. It is the first study of its kind and the most comprehensive to date in terms of the time-windows investigated. Before this study we did not have sufficient knowledge to select, in an informed manner, the full range of regions of interest that might show oxytocin-induced changes across all these methods of administration, the length of the temporal window, and how to model the temporal dynamics of the brain effects. Thanks to this study we now do. While our analytic approach had certain benefits and the use of *a priori* statistical criteria is essential (we cannot be adjusting the number of time intervals included and the thresholds until we observe an effect), they presented the danger of either reporting fragmented pharmacodynamics effects or not observing an effect that may have been just sort of the threshold, leading to possible paradoxes, as highlighted by the

reviewer. We believe that the new analysis we present helped address these concerns, generate testable hypotheses regarding the spatial and temporal dynamics of oxytocin-induced changes in rCBF, by method of administration, and inform future studies aiming to target specific anatomical regions by providing an indication of the time intervals post-dosing at which these effects may be present and/or maximal for each region.

Finally, we would just like to note that the choice of time interval to implement a behavioural intervention post intranasal oxytocin dosing, to which the reviewer refers, has rather been arbitrary and driven by historical precedence and not informed by a comprehensive investigation of the time-course of the pharmacodynamics of intranasal oxytocin. We are aware of only two studies where the temporal dynamics of the effects of intranasal oxytocin on human brain function have been examined. The first is the previous validation study from our group: a resting state, *single-blind*, *between-subjects* study where we investigated the effects of a dose of 40 IU or placebo 25-78 mins post-dosing (compared to *baseline, not placebo*), reporting, using pattern recognition, a pharmacodynamics profile consistent with a peak at 39-51 mins post-dosing(1). The second is a study where the authors investigated the temporal effects of a dose of 24 IU on amygdala reactivity to fearful faces in three time intervals: 15, 45 and 75 mins, using BOLD fMRI. This study cannot inform on potential effects beyond the circuits engaged by this specific task, the specific time-points sampled or the limitations of BOLD-fMRI used to investigate amygdala responses (2). For their own shortcomings, neither study could fully inform us on the spatial and temporal dynamics of intranasal oxytocin, and of course both studies used standard nasal sprays.

The administration of the huge amount of OT may saturate potential nose-to-brain pathway (if existing). Can the authors comment on this issue?

The reviewer raises a very interesting question for debate here: given the high amount of oxytocin typically administered intranasally, shouldn't we expect some degree of saturation of the potential nose-to-brain pathways, if they exist? If this saturation occurs, then one would expect that the effects of spray and nebuliser would converge, given that any extra-deposition of oxytocin at the target regions with the nebuliser would not be able to impact the amounts of oxytocin potentially transported centrally. While this is an empirical question and no study to date have explored this question, we believe some further quantitative considerations should be made about the potential of spray versus nebuliser to target the olfactory/paranasal sinuses.

It is important to consider that nasal sprays are not optimized to deliver molecules beyond the nasal valve, which means the actual targeting of superior/posterior areas of the nasal activity, where for instance the olfactory region is located and the nose-to-brain transport might occur, is extremely low (according to one study using biophysical models of the nasal cavity < 4.6 % is deposited in the olfactory region (3); in another study using scintigraphy even lower deposition rates were observed – 2.9±1.8%) (4). Increases in the deposition in the olfactory region of up to 15.7 (±2.4) % of the administered volume can be achieved with the PARIS sinus nebuliser (5). This means that at least 10% more oxytocin can actually reach the olfactory region using the nebuliser. It is an empirical question whether this small deposition of < 4.6% of the administered dose with the spray can already saturate the pathway and we are not aware of any evidence supporting or disproving this idea. However, there is also another crucial difference between nasal sprays and the nebuliser tested herein, which is the capacity of the nebuliser to reach areas that the spray is less likely to target, namely the paranasal sinuses. The paranasal sinuses have a much higher surface area of deposition than the olfactory region and are barely targeted by nasal sprays (<1% of spray

reaches the sinuses(4)). The delivery to the paranasal sinuses seems to be increased about 3-5x by using the aerosol pulsation deposition method implemented by the PARI sinus nebuliser(6, 7). The paranasal sinus mucosa is also innervated by the trigeminal nerve which is one of the pathways proposed to underlie nose-to-brain transport (8). Therefore, one may argue that if the concentrations deposited in the olfactory region are unlikely to saturate the pathway of transport to the brain (if it exists), they are even less likely to do so in the mucosa of the paranasal sinuses innervated by the trigeminal pathway. It is therefore plausible that in the absence of saturation of the pathway of transport these increases of 3-5 times in deposition on target areas (such as the olfactory region or the paranasal sinuses) when using the nebuliser may result in increased central bioavailability of oxytocin, especially when we could not detect any differences in the amounts of oxytocin absorbed into the peripheral circulation or dripped down to the buccal cavity, as estimated by measuring oxytocin in saliva post-dosing (unpublished data). Such increased bioavailability would match well the unique pattern of changes we observe for the nebuliser, namely increases in rCBF in areas of the brain distal from the putative point-of-entry. As we acknowledge in our discussion (page 15) this interpretation is speculative. Ideally, this question should be followed up with positron imaging tracer studies, where the amounts of tracer reaching the brain could be compared between spray and nebuliser directly. However, this is not yet feasible for two reasons: 1) first, no valid tracer is available for oxytocin; 2) second, intranasal administration of radiotracers may be associated with unacceptable exposure to radiation in humans, as suggested by a recent simulation study departing from data acquired in rodents(9). Studies in rodents, such as the one mentioned by the reviewer in the next point - *Smith et al. 2019*, could also bring some insights to answer this empirical question. Nevertheless, these methods of administration (spray and nebuliser) are not available for administration in rodents (and for sure the topical direct applications of the peptide in the olfactory mucosa typically used in intranasal administrations in rodents is unlikely to resemble the deposition achieved when a nasal spray or nebuliser is used in humans).

In summary, no empirical data is available to support or reject the idea that the nose-to-brain pathway of transport can be saturated and to inform us about the amount of administered peptide that may achieve saturation. Nevertheless, this aspect reflects a critical assumption regarding the interpretation of our findings for the nebuliser, which we now explicitly acknowledge (page 15, paragraph 2): *"This hypothesis is mostly valid assuming that the amounts delivered with the spray are not enough to saturate the pathway of transport from the nose to the brain, if it exists."*

3) The author may cite and probably compare their results with recently published results from Scott Young lab: Smith et al. "Oxytocin delivered nasally or intraperitoneally reaches the brain and plasma of normal and oxytocin knockout mice". Pharmacology research, 2019.

We thank the reviewer for the suggestion. We also agree the results presented in this study are compelling and lend support to our observations in humans. When administered intraperitoneally, synthetic oxytocin reaches the extracellular fluid in the amygdala within 30 mins after administration and its concentration remains elevated for about 1h. This is very much in agreement with what we found for the effects of intravenous administration of synthetic oxytocin on rCBF in the left amygdala. At the same time, the authors also report that the intranasal administration of synthetic oxytocin not only seems to reach the extracellular fluid in the brain but may also come with benefits (although small) in terms of achieving higher concentrations in the extracellular fluid in the brain. This aspect is in line with our findings for the intranasal route in humans, where the intranasal route (either spray or nebuliser) seems to result in changes in rCBF that cannot be echoed by intravenous oxytocin and that we believe may reflect these

small increases in central bioavailability after intranasal administrations (that may certainly sum up to the amounts derived from the peripheral circulation). We have now included the data presented in this manuscript in our revised version of the discussion (Page 14).

Reviewer #3 (Remarks to the Author):

The authors have responded carefully and appropriately to all of the issues that I raised, and as far as I can see to the other points also.

There are certainly statements in this paper that I would disagree with and I remain skeptical about some of the interpretation. However, this study is by the standards of the field, exceptionally rigorous, carefully controlled and the interpretation balanced and thoughtful.

Including intravenous oxytocin infusions as a control is an important and practical step to include in future behavioral and brain imaging studies using oxytocin: intravenous delivery will be more reliable, controllable and reproducible than intranasal delivery, and clearly there is at least very considerable overlap in its consequences for brain activity. However it should be much easier using IV delivery to establish exactly where the primary sites of action are _ including the key question of whether these are peripheral or central.

We thank the reviewer for his appraisal of our work as exceptionally rigorous and scholarly. We fully share the reviewer's view regarding the importance of peripheral comparators in oxytocin studies and the potential our findings hold to prompt further investigation using this alternative method of administration, which as the reviewer highlights, comes with many advantages.

Reviewer #4 (Remarks to the Author):

The article by Martins et al "Do direct nose-to-brain pathways underlie intranasal oxytocin-induced changes in regional cerebral blood flow in humans?" tackles a controversial issue and has triggered some controversial reviews. In reading the article and the accompanying comments, I will only add in comments as they relate to my areas of competency: perfusion imaging and functional MRI analysis. I remain agnostic to the argument for separate pathways between the nose and the brain, circumventing the BBB. I hope my comments will be useful to the authors for a re-analysis and a stronger resubmission of this article, which tackles an interesting and innovative question.

Minor comments:

P5: "...As a result of neuro-vascular coupling, changes in rCBF are likely to reflect changes in neuronal activity rather than vascular effects..." by definition, rCBF methods would have to capture vascular effects as well as neural activity. Remove the clause "rather than vascular effects"

We agree with the reviewer here and have removed the clause "rather than vascular effects" as suggested (page 5).

Figure 4: the legend needs another caption/line denoting the color used for the overlapping of regions.

Thanks for the suggestion. We agree including this another caption makes the figure easier to interpret. This caption has been included in our now revised Fig. 4.

Page 15: The sentences “... Our initial hypothesis was that the nebuliser would result in a more robust pattern of changes that would be comparable to the spray but of higher magnitude and eventually include areas that could not be targeted neither by the spray nor by the intravenous administrations. However, we should acknowledge that this prediction would have mostly been valid if the pharmacodynamics of the rCBF response to OT followed a linear model – which does not seem to be the case at least for some brain areas...” seem like run-on sentences and are very difficult to follow. Please edit accordingly.

We have now edited the sentences to improve the flow and clarity. Thanks for the suggestion. The sentence now reads as below:

“Our initial hypothesis was that the nebuliser would result in changes similar to those achieved by the spray but of higher magnitude. We also considered that the nebuliser might modulate rCBF in areas that were not targeted either by the spray or the intravenous administrations.” (Page 16)

Major comments:

The strategy of analysing the data a large number of t-tests seems like a poor use of the data and invites problems with multiple comparisons, as discussed. Accordingly, this approach yields a temporal rCBF evolution that doesn't make a lot of sense, as pointed out by another of the reviewers.

Given that there is no a priori model for the temporal evolution of the rCBF, there are better options. They could easily use PCA or ICA to search for patterns in the time course.

We completely understand the reviewer's concerns, which were also shared by reviewer 2. Thanks to both reviewers' concerns, ideas and encouragement, we have not undertaken a new analysis that makes fuller justice to our data. Please see our response to comments 1&2 of Reviewer 2 where we provide a full justification of our approach, acknowledge its limitations, and describe the new analysis that can inform on the temporal dynamics of oxytocin-induced changes in rCBF in the context of what was feasible given the already undertaken analyses (i.e. avoiding circularity).

We also completely agree with the reviewer that in the absence of an *a priori* model for the temporal evolution of the effects of oxytocin on rCBF, an exploratory data-driven approach, such as principal component analysis or independent component analysis, might be useful to explore the temporal dynamics of regional effects in an unbiased manner. In fact, we had already tasked an *MSc* student to conduct these analyses for his thesis. However, this exploratory data analytic approach did not yield meaningful information regarding the temporal dynamics of oxytocin effects. We believe this may be due to the small number of data points (only 40) for such data-driven analyses (compared to the hundreds of time-points typically used in resting-state BOLD fMRI). While interesting for our own exploration of the

data, we believe it would only add confusion to include these results in the current manuscript. If the reviewer is interested though, we would be happy to share the *MSc* thesis with him/her.

Additionally, I think it is crucial to extract the rCBF time courses (8 time points) from preconstructed ROIs and try to generate a kinetic model based on those. The plasma concentration of OT should be part of the model. I note that the 8 minute ASL acquisitions had many averages. For the purpose of time series analysis and kinetic modelling, it is more efficient to NOT average those images, but rather use a larger number of noisier time points. The authors have 5 pairs of images collected 8 times. Using a surround subtraction scheme, they could be 38 time points instead.

We thank the reviewer for his/her suggestions. Some of these suggestions inspired us to conduct the new analysis we described in response to comments 1&2 by reviewer 2, which indeed offers a better insight into the temporal dynamics of the effects of oxytocin on rCBF by brain region and method.

Regarding the suggestion to create a pharmacokinetic/pharmacodynamics (pK/PD) model to describe the effects of synthetic oxytocin on rCBF, incorporating the concentration of oxytocin in plasma post-dosing: we took the reviewer's suggestion to the heart and discussed this possibility further with experts in pK/pD modelling to examine whether this approach could be a feasible option in our specific case. While this idea is interesting from a theoretical point of view, the consensus was that these analyses would not be feasible to implement in our case for three main reasons:

- (1) Since we relied on intranasal administrations where absorption to plasma *and* putative direct transport to the brain may co-occur, it is not obvious what input function to use for the nasal delivery. The fractions of administered dose that cross to the brain using the nose-to-brain pathway are currently unknown for humans. Additionally, we cannot extrapolate from rodent studies due to the pronounced species differences in nasal anatomy (10).
- (2) Since we are not measuring the amount of oxytocin reaching the extracellular brain fluid after administration with the various methods (this would be virtually impossible humans), we also do not have a compelling way of linking local pharmacokinetics with downstream pharmacodynamics effects such as the changes in rCBF that we did measure. Assuming homogenous and equal distribution in the brain is likely to be a wrong assumption in intranasal administrations, given that rodent studies suggest bioavailability may vary as a function of the point of entry (11).
- (3) We do not have a clear idea (or at least a fair approximation) of the density of the oxytocin receptors in each region and of the local concentration of peptide that needs to be achieved to engage changes in rCBF at each region. According to what we already know about the pharmacology of the oxytocin system this would be virtually impossible to approximate, since activation of the oxytocin receptor may engage different intracellular pathways depending on the amount of peptide available. For instance, whether oxytocin recruits G_s stimulatory versus G_i inhibitory proteins depends on its extracellular concentration (12).

In summary, we do not believe that we currently have the information we need to build a compelling pK/PD model that would add something to the manuscript. Nevertheless, if the reviewer has suggestions on how to overcome these issues, we are happy to revisit this analysis in a revised version of the manuscript.

What is the correlation between plasma OT and rCBF? Those correlation maps would be very helpful in settling the debate.

The reviewer raises an interesting point. Regarding these correlations between plasma oxytocin and rCBF, we decided to approach this in a more hypothesis-driven rather than exploratory approach. We estimated the correlations between the first eigenvariate component reflecting the effect of oxytocin on rCBF for each cluster where we found significant treatment effects in our paired-T tests analyses (adjusting for global CBF as a covariate of no interest) and the area under the curve reflecting individual differences in treatment-induced increases in oxytocin plasma concentrations over time, for the corresponding method of administration. We found that plasma oxytocin could only predict decreases in rCBF for the clusters where we found an overlap between the intravenous and spray administrations, but not for the other clusters where spray and nebuliser modulated rCBF, but intravenous oxytocin had no effect. We describe these analyses in the Materials and Methods section (page 32) and present the results in the Results section (page 9), Fig. 5 and Supplementary table S3.

Page 16: The authors argue "... If direct modulation of activity in these nerve bundles would account for all the changes we report for our intranasal administration methods, then one would expect that the changes observed in rCBF after intranasal administration of synthetic OT would be mostly restricted to the olfactory/trigeminal pathways and respective connected areas, which is not entirely supported by our data (for example, we have little evidence to believe that modulation of activity in olfactory or trigeminal nerves would result in changes in rCBF in the frontal gyrus)... ". It is just as likely that the authors have detected an unexpected connectivity pathway as that they have detected a nose-brain transport system.

The reviewer raises a valid point. Nevertheless, we would like to highlight that studies using trigeminal nerve stimulation in healthy individuals have been quite consistent in suggesting that the stimulation of the trigeminal nerve is more likely to impact on brainstem polysynaptic circuits rather than on cortical excitability (whether the stimulation is acute or chronic) (13, 14). Combined with the fact that most of the reported oxytocin-induced changes in rCBF map on those brain areas that we currently know to express receptors sensitive to oxytocin (either the OTR or the AVPR1A) in the human brain, this evidence would support the interpretation that local changes in the activity of the olfactory or trigeminal nerve bundles are unlikely to fully account for all the changes we report after the intranasal administration of oxytocin. Nevertheless, we cannot exclude they may have some minor contribution. We discuss this topic on pages 17 and 18.

We would also like to note that we never claimed that we detected a nose-to-brain pathway (this was not our aim in any case), but simply that our data, given our current understanding of this system, are consistent with the existence of nose-to-brain pathways.

The authors are careful to remove global CBF effects. The global CBF time course may contain valuable information about the effect of OT and accounts for the differences between administration routes and would deserve its own analysis as well as serve as a regressor of no interest.

We totally agree with the reviewer and shared the same concern. In fact, we had already presented in the previous form of the manuscript the analyses for main effects of time, treatment and time x treatment interaction on global CBF. These data are presented in the results section of the main manuscript (page 6, section 1) and supplementary material (Fig. S1, including time-courses for each level of the treatment factor). We found no significant effects of treatment or interaction treatment x time interval. As expected, the main effect of time interval was significant (something that is typically described in perfusion studies over long periods of time and is likely to reflect small drops in heart-rate or blood pressure after long periods of time in the supine position(15)). Even though the treatment or treatment x time interval effects were not significant, the visual inspection of Fig. S1 showed nominal decreases in global CBF following the intranasal administration of oxytocin, particularly at early time intervals with the nasal spray. For this reason, we conservatively included global CBF as a regressor of no interest in all of our analyses.

References

1. Paloyelis Y, Doyle OM, Zelaya FO, Maltezos S, Williams SC, Fotopoulou A, et al. A Spatiotemporal Profile of In Vivo Cerebral Blood Flow Changes Following Intranasal Oxytocin in Humans. *Biological Psychiatry*. 2016;79(8):693-705.
2. Spengler FB, Schultz J, Scheele D, Essel M, Maier W, Heinrichs M, et al. Kinetics and Dose Dependency of Intranasal Oxytocin Effects on Amygdala Reactivity. *Biological Psychiatry*. 2017;82(12):885-94.
3. Cheng YS, Holmes TD, Gao J, Guilmette RA, Li S, Surakitbanharn Y, et al. Characterization of nasal spray pumps and deposition pattern in a replica of the human nasal airway. *J Aerosol Med*. 2001;14(2):267-80.
4. Djupesland PG, Skretting A. Nasal Deposition and Clearance in Man: Comparison of a Bidirectional Powder Device and a Traditional Liquid Spray Pump. *J Aerosol Med Pulm D*. 2012;25(5):280-9.
5. Xi JX, Yuan JE, Zhang Y, Nevorski D, Wang ZX, Zhou Y. Visualization and Quantification of Nasal and Olfactory Deposition in a Sectional Adult Nasal Airway Cast. *Pharmaceutical Research*. 2016;33(6):1527-41.
6. Moller W, Schuschnig U, Saba GK, Meyer G, Junge-Hulsing B, Keller M, et al. Pulsating aerosols for drug delivery to the sinuses in healthy volunteers. *Otolaryng Head Neck*. 2010;142(3):382-8.
7. Xi JX, Wang ZX, Nevorski D, White T, Zhou Y. Nasal and Olfactory Deposition with Normal and Bidirectional Intranasal Delivery Techniques: In Vitro Tests and Numerical Simulations. *J Aerosol Med Pulm D*. 2017;30(2):118-31.
8. Pardeshi CV, Belgamwar VS. Direct nose to brain drug delivery via integrated nerve pathways bypassing the blood-brain barrier: an excellent platform for brain targeting. *Expert Opinion on Drug Delivery*. 2013;10(7):957-72.
9. Beard R, Singh N, Grundschober C, Gee AD, Tate EW. High-yielding F-18 radiosynthesis of a novel oxytocin receptor tracer, a probe for nose-to-brain oxytocin uptake in vivo. *Chem Commun*. 2018;54(58):8120-3.
10. Reznik GK. Comparative Anatomy, Physiology, and Function of the Upper Respiratory-Tract. *Environmental Health Perspectives*. 1990;85:171-6.
11. Smith AS, Korgan AC, Young WS. Oxytocin delivered nasally or intraperitoneally reaches the brain and plasma of normal and oxytocin knockout mice. *Pharmacological Research*. 2019;146.

12. Manning M, Misicka A, Olma A, Bankowski K, Stoev S, Chini B, et al. Oxytocin and Vasopressin Agonists and Antagonists as Research Tools and Potential Therapeutics. *Journal of Neuroendocrinology*. 2012;24(4):609-28.
13. Mercante B, Pilurzi G, Ginatempo F, Manca A, Follesa P, Tolu E, et al. Trigeminal nerve stimulation modulates brainstem more than cortical excitability in healthy humans. *Experimental brain research*. 2015;233(11):3301-11.
14. Axelson HW, Isberg M, Flink R, Amandusson A. Trigeminal Nerve Stimulation Does Not Acutely Affect Cortical Excitability in Healthy Subjects. *Brain Stimul*. 2014;7(4):613-7.
15. Howard MA, Krause K, Khawaja N, Massat N, Zelaya F, Schumann G, et al. Beyond patient reported pain: perfusion magnetic resonance imaging demonstrates reproducible cerebral representation of ongoing post-surgical pain. *PLoS One*. 2011;6(2):e17096.

Reviewers' Comments:

Reviewer #2:

Remarks to the Author:

The manuscript has been significantly improved and deserves publication.

Reviewer #3:

Remarks to the Author:

It seems to me that the study is methodologically sound, and the conclusions justified. How important this paper is a matter of judgement. There have been a great many papers published using intranasal oxytocin but almost none have controlled, as this study has attempted to, for the consequences of intranasal application of very large doses of oxytocin on peripheral concentrations of oxytocin.

Referee 1 is (in my view) correct in that this paper does little to elucidate the mechanisms of action of intranasal oxytocin on the brain, but it seems to me that the authors have made a good case that (1) increases in plasma oxytocin induce certain changes in brain perfusion ; but (2) these increases cannot account for all changes in brain perfusion observed following the intranasal administration of oxytocin.

The novelty of this manuscript rests on the careful attention paid to understanding the contribution of changes in circulating oxytocin to changes observed in brain activity. It is true that some of the effects of high circulating oxytocin concentrations might be the result of passage of some oxytocin across the blood-brain barrier; equally it is possible that differences between the routes might reflect differences in the dynamics of peripheral changes.

Thus there is residual uncertainty of interpretation. Nevertheless, I would reiterate that by the standards of the field this is an exceptionally rigorous study, and one that provides an important indication that in studies using intranasal oxytocin, the consequences of the ensuing high peripheral concentrations of oxytocin must be considered.

Reviewer #4:

Remarks to the Author:

the authors have satisfactorily answered my concerns.

Reviewers' comments:

Reviewer #2 (Remarks to the Author):

The manuscript has been significantly improved and deserves publication.

We would like to thank the reviewer for his/her time and insightful and constructive comments, which we believe helped improve our manuscript substantially.

Reviewer #3 (Remarks to the Author):

It seems to me that the study is methodologically sound, and the conclusions justified. How important this paper is a matter of judgement. There have been a great many papers published using intranasal oxytocin but almost none have controlled, as this study has attempted to, for the consequences of intranasal application of very large doses of oxytocin on peripheral concentrations of oxytocin.

Referee 1 is (in my view) correct in that this paper does little to elucidate the mechanisms of action of intranasal oxytocin on the brain, but it seems to me that the authors have made a good case that (1) increases in plasmatic oxytocin induce certain changes in brain perfusion ; but (2) these increases cannot account for all changes in brain perfusion observed following the intranasal administration in oxytocin.

The novelty of this manuscript rests on the careful attention paid to understanding the contribution of changes in circulating oxytocin to changes observed in brain activity. It is true that some of the effects of high circulating oxytocin concentrations might be the result of passage of some oxytocin across the blood-brain barrier; equally it is possible that differences between the routes might reflect differences in the dynamics of peripheral changes.

Thus there is residual uncertainty of interpretation. Nevertheless, I would reiterate that by the standards of the field this is an exceptionally rigorous study, and one that provides an important indication that in

studies using intranasal oxytocin, the consequences of the ensuing high peripheral concentrations of oxytocin must be considered.

We would like to thank the reviewer for their time, their insightful comments and their balanced appraisal of our work.

Reviewer #4 (Remarks to the Author):

the authors have satisfactorily answered my concerns.

We would like to thank the reviewer for his/her time and insightful and constructive suggestions for further analyses, which highlighted important aspects of our data and helped improve our manuscript substantially.